# Trend analysis of the airborne fraction and sink rate of anthropogenically released $CO_2$

Mikkel Bennedsen[1,3], Eric Hillebrand[1,3], and Siem Jan Koopman[2,3]

[1]Department of Economics and Business Economics, Aarhus University, Fuglesangs Allé, 4 8210 Aarhus V, Denmark
[2]Department of Econometrics, School of Business and Economics, Vrije Universiteit Amsterdam, De Boelelaan 1105, 1081 HV Amsterdam, The Netherlands.
[3]Center for Research in Econometric Analysis of Time Series (CREATES), Aarhus University, Fuglesangs Allé, 4 8210 Aarhus V, Denmark

**Correspondence:** Mikkel Bennedsen (mbennedsen@econ.au.dk)

**Abstract.** Is the fraction of anthropogenically released $CO_2$ that remains in the atmosphere (the airborne fraction) increasing? Is the rate at which the ocean and land sinks take up $CO_2$ from the atmosphere decreasing? We analyze these questions by means of a statistical dynamic multivariate model, from which we estimate the unobserved trend processes together with the parameters that govern them. We show how the concept of a global carbon budget can be used to obtain two separate data series measuring the same physical object of interest, such as the airborne fraction. Incorporating these additional data into the dynamic multivariate model increases the number of available observations, thus improving the reliability of trend and parameter estimates. We find no statistical evidence of an increasing airborne fraction but we do find statistical evidence of a decreasing sink rate. We infer that the efficiency of the sinks in absorbing $CO_2$ from the atmosphere is decreasing at approximately $0.54\%$ per year.

## 1 Introduction

A part of the anthropogenically released $CO_2$ emitted to the atmosphere flows to the oceans (the ocean sink) and the terrestrial biosphere (the land sink). Approximately $45\%$ of released $CO_2$ stays in the atmosphere (the airborne fraction), while the two sinks take up approximately $24\%$ and $31\%$ of the $CO_2$, respectively. (These percentages are calculated over the period 1959 to 2016 using the data described below, see e.g. Raupach et al., 2014, for similar estimates.) A key question is whether the airborne fraction is increasing or if it remains constant at around $45\%$. An increasing airborne fraction implies that the share of anthropogenically released $CO_2$ that ultimately remains in the atmosphere increases, and projections of future atmospheric $CO_2$ levels need to take this into account (Gloor et al., 2010). Closely related is the question whether the sinks will continue taking up $CO_2$ at the same rate (the sink rate) or if this rate is decreasing. A decreasing sink rate implies that the efficiency with which ocean and land sinks are absorbing $CO_2$ from the atmosphere is decreasing. Thus, analyzing the behavior of the sink

rate can help predict the future uptake of $CO_2$ through the ocean and the land sink. The answers to the questions posed above are important for our understanding of the global carbon cycle and are relevant for policy makers and the public in general.

A series of papers argue that the airborne fraction of anthropogenically released $CO_2$ (mainly through fossil fuel emissions, cement production, and land-use change) is increasing (Canadell et al., 2007a; Le Quéré et al., 2009; Raupach et al., 2008; Rayner et al., 2015). Similarly, in Raupach et al. (2014) it is argued that, although the statistical evidence of an increasing airborne fraction is relatively weak, the evidence of a decreasing $CO_2$ sink rate is clearer. However, the methods in these studies have been criticized in, for example, Knorr (2009), Gloor et al. (2010), and Ballantyne et al. (2015). Indeed, by considering a longer data set and incorporating uncertainties into the data, Knorr (2009) found that the conclusion of an increasing airborne fraction was not warranted. Similarly, Ballantyne et al. (2015) argues that errors in the data can lead to erroneous conclusions regarding possible trends in the airborne fraction and in the sink rate.

In this paper, we address these statistical issues within the framework of a *state-space system*. It allows us to conduct statistical inference by taking explicit account of stochastic and deterministic trends in the data, transient shocks to the data (coming from, e.g., volcanic eruptions or strong El Niño events), and (potential) measurement errors. It also allows for the simultaneous incorporation of multiple data sets for the same object, which can improve the estimation of trends and increase reliability of parameter estimates. We find strong evidence for purely deterministic trends when we incorporate multiple measurements for the airborne fraction and the sink rate. These deterministic trends have a statistically significantly negative slope in the case of the sink rate and an insignificant slope in the case of the airborne fraction. These findings corroborate earlier findings in the literature, especially Raupach et al. (2014), but address the statistical concerns raised by Knorr (2009) and Ballantyne et al. (2015), among others. Finally, by analyzing the ocean and land sink rates separately, we find no evidence of a decreasing ocean sink rate but we do find evidence that the land sink rate is decreasing.

The paper is organized as follows. In Sect. 2 we state the fundamental equations of the global carbon budget, the definitions of the airborne fraction of anthropogenically released $CO_2$, and the $CO_2$ sink rate, which will motivate the specification of the state-space system. Sect. 3 introduces the state-space system used in the paper. In Sect. 4 we conduct a trend analysis of the airborne fraction within the proposed statistical framework. In Sect. 5 we carry out the corresponding analysis of the $CO_2$ sink rate, and in Sect. 6 of the land and ocean sink rates separately. Sect. 7 discusses the results and Sect. 8 concludes. A Supplementary Material file is available online.

## 2 The global carbon budget

The so-called *global carbon budget* is defined as

$$E_t^{ANT} = G_t + S_t^O + S_t^L, \tag{1}$$

where $E_t^{ANT}$ is anthropogenically released $CO_2$ into the atmosphere, $G_t$ is growth of atmospheric $CO_2$ concentration, $S_t^O$ is the flux of $CO_2$ from the atmosphere to the oceans (the ocean sink), and $S_t^L$ is the flux of $CO_2$ from the atmosphere to the terrestrial biosphere (the land sink). In words, Eq. (1) states that emissions of $CO_2$ should equal the fluxes of $CO_2$ to the

atmosphere, the ocean sink, and the land sink. We use the data set provided by The Global Carbon Project (Le Quéré et al., 2018).[1] All data are given in gigatonnes of carbon (GtC) and are recorded at a yearly frequency, beginning in 1959 and ending in 2016, resulting in 58 observations for each quantity in (1).

While the carbon budget is in principle always balanced for the physical quantities, in the sense that Eq. (1) always holds,
this might not be the case when inserting actual data for emissions and sinks, due to measurement errors in the data. For this reason, Le Quéré et al. (2018) introduce a residual term into the budget Eq. (1) to capture measurement error. It is denoted $B_t^{IM}$ for *budget imbalance*. Therefore, when considering actual data, the carbon budget is defined as

$$E_t^{ANT} = G_t + S_t^O + S_t^L + B_t^{IM}. \tag{2}$$

The sample mean of the budget imbalance over the observation period is not significantly different from zero and shows no
sign of a trend (Le Quéré et al., 2018). These facts are important in the developments below, since they motivate treating $B_t^{IM}$ as part of an error term.

The growth rate in atmospheric $CO_2$ data, $G_t$, is from Dlugokencky and Tans (2018), the ocean sink data, $S_t^O$, are obtained from an ensemble of global biochemistry models, and the land sink data, $S_t^L$, are estimated as the multi-model mean of several dynamic global vegetation models. See Le Quéré et al. (2018) for further information on the data. The anthropogenic emissions
of $CO_2$ can be decomposed in two parts:

$$E_t^{ANT} = E_t^{FF} + E_t^{LUC},$$

where $E_t^{FF}$ are emissions from fossil fuel burning, cement production, and gas flaring, while $E_t^{LUC}$ are emissions from land-use change. Fossil fuel emissions, $E_t^{FF}$, are from Boden et al. (2018), while land-use change emissions, $E_t^{LUC}$, are averages of the results of the two bookkeeping models of Hansis et al. (2015) and Houghton and Nassikas (2017), updated as in Le Quéré
et al. (2018). The time series of concentrations (above preindustrial levels) of $CO_2$ in the atmosphere is constructed as

$$C_t = 2.127 \cdot ([CO_2]_{1959} - [CO_2]_{1750}) + \sum_{\tau=1}^t G_\tau,$$

where $[CO_2]_{1750} = 279$ ppmv (parts per million volume) and $[CO_2]_{1959} = 315.39$ ppmv are the concentrations of $CO_2$ in the atmosphere in 1750 and 1959, respectively; see Raupach et al. (2014). The number 2.127 is the conversion factor from ppmv to GtC. In words, the atmospheric concentration $C_t$ above pre-industrial levels is given by the initial value in 1959 plus the
cumulative sum of the growth in atmospheric $CO_2$ concentrations $G_t$, which result from the budget equation (1).

We follow Raupach (2013) and Raupach et al. (2014) and define the airborne fraction as

$$AF_t = \frac{G_t}{E_t^{ANT}}$$

and the $CO_2$ sink rate as

$$k_{S,t} = \frac{S_t^O + S_t^L}{C_t}, \tag{3}$$

---

[1]The data are available at http://www.globalcarbonproject.org/ and were downloaded on June 1st, 2018.

which is the flux of $CO_2$ from the atmosphere to the sinks (ocean plus land), normalised by the amount of $CO_2$ (above preindustrial levels) currently in the atmosphere. We can also consider the individual components of the sink rate for ocean and land, which are given by

$$k_{O,t} = \frac{S_t^O}{C_t}, \qquad k_{L,t} = \frac{S_t^L}{C_t}, \tag{4}$$

respectively, with $k_{S,t} = k_{O,t} + k_{L,t}$.

The airborne fraction and the sink rate are fundamentally different quantities. The airborne fraction $AF_t = G_t/E_t^{ANT}$ is the ratio of the growth of atmospheric $CO_2$ in period $t$ to the amount of $CO_2$ emitted in period $t$. It is thus a measure of the fraction of emitted $CO_2$ that stays in the atmosphere. In contrast, the sink rate $k_{S,t} = (S_t^O + S_t^L)/C_t$ is the ratio of the $CO_2$ flux in the sinks in period $t$ to the total amount of $CO_2$ in the atmosphere (above pre-industrial levels). By writing $S_t^O + S_t^L = k_{S,t}C_t$, we

can interpret the sink rate $k_{S,t}$ as the "efficiency", with which $CO_2$ flows from the atmosphere to the sinks, i.e. as the amount of $CO_2$ going into the sinks for an extra unit of $CO_2$ added to the atmosphere (Gloor et al., 2010; Raupach, 2013). We discuss the relationship between the airborne fraction and the sink rate further in Sect. 7.

## 3   Trend model specification

In this section, we consider several models for the data generating process behind observations of the objects of interest defined

in Sect. 2. Common to all models is that they can be cast in a state-space system of the form:

$$\begin{aligned} y_t &= A\,x_t + \xi_t, \\ x_{t+1} &= B\,x_t + \kappa_t, \end{aligned} \qquad t = 1, \ldots, n, \tag{5}$$

where $y_t$ is a vector of observations at time $t = 1, \ldots, n$ with time series length $n$, and the system matrices $A$ and $B$ have appropriate dimensions. The vector $x_t$ is usually referred to as the state vector, which can include deterministic and stochastic trends, and the error terms $\xi_t$ and $\kappa_t$ are both independent and identically distributed (iid) random vectors of appropriate

dimension and with mean zero. For example, when we need to model the airborne fraction alone, we have $y_t = AF_t$ and the state-space system represents a univariate dynamic model for the airborne fraction. When modelling the ocean and land sink rates jointly, we have $y_t = (k_{O,t}, k_{L,t})'$, and the state-space system is a bivariate dynamic model. For given matrices $A$ and $B$, and under the assumption of mutually and serially uncorrelated Gaussian errors $\xi_t$ and $\kappa_t$ (with their respective variance matrices $\Sigma_\xi$ and $\Sigma_\kappa$), the state-space system is a linear Gaussian model. In such regular cases, an analytic formulation for the

likelihood function is available and relies on the prediction error decomposition. Hence the parameters (variances and possibly covariances in $\Sigma_\xi$ and $\Sigma_\kappa$) can be estimated by the maximum likelihood method. It requires the numerical optimization of the log-likelihood function that is evaluated via the Kalman filter. The resulting algorithm is initialized with specific starting values; we use a diffuse initialization as outlined in Chapter 5 of Durbin and Koopman (2012). The smooth estimate of the state process $x_t$ can also be obtained by means of the Kalman filter together with a smoothing algorithm. The extracted state is

effectively the conditional mean $\mathbb{E}(x_t|y_1, \ldots, y_n; A, B, \Sigma_\xi, \Sigma_\kappa)$, for $t = 1, \ldots, n$. Details of the state-space approach are given by Durbin and Koopman (2012), where both signal extraction and maximum likelihood estimation are discussed.

Our baseline model is the local linear trend (LLT) model. For a univariate time series $y_t$, we treat the underlying trend $T_t$ as a stochastic process given by

$$T_{t+1} = T_t + \beta + \eta_t, \tag{6}$$

where $\beta \in \mathbb{R}$ is a fixed and unknown coefficient and $\eta_t$ is an iid Gaussian random variable with mean zero and variance $\sigma_\eta^2$. The solution to the difference equation (6) is given as

$$T_{t+1} = T_1 + t\beta + \sum_{i=0}^{t-1} \eta_{t-i}, \qquad t = 1, 2, \ldots, n-1,$$

where $T_1$ can be treated as a fixed unknown coefficient (intercept or constant) or as a random variable. The solution shows that the trend component is made up of the starting value $T_1$, a deterministic linear term with slope $\beta$, and a random walk component $\sum_{i=0}^{t-1} \eta_{t-i}$. Thus, $T_t$ can be interpreted as a long-term trend in the time series and $\beta$ as the slope of the deterministic part of the trend. We also considered a time-varying slope, $\beta_t$, but found no evidence supporting this generalization in either the airborne fraction or the sink rate. The observation equation for $y_t$ is given by

$$y_t = T_t + \epsilon_t, \tag{7}$$

where $T_t$ is given by (6) and $\epsilon_t$ captures deviations of the observed time series from the unobserved trend component. The deviations $\epsilon_t$ can be viewed as $(i)$ actual (transient) disturbances of the physical systems arising from, for example, volcanic eruptions and El Niño events, and/or $(ii)$ measurement errors arising from the way the data are collected. The random variable $\epsilon_t$ is assumed to be iid Gaussian with mean zero and variance $\sigma_\epsilon^2$.

The local linear trend model can be cast in the state-space system (5) where vectors and matrices are defined as

$$x_t = \begin{pmatrix} T_t \\ \beta \end{pmatrix}, \quad A = \begin{bmatrix} 1 & 0 \end{bmatrix}, \quad B = \begin{bmatrix} 1 & 1 \\ 0 & 1 \end{bmatrix}, \quad \xi_t = \epsilon_t, \quad \kappa_t = \begin{pmatrix} \eta_t \\ 0 \end{pmatrix},$$

for $t = 1, \ldots, n$. The state vector $x_t$ consists of the two variables of interest: stochastic trend variable $T_t$ and deterministic slope variable $\beta$. The state-space methods as discussed above can treat such mixed compositions of the state vector. We have illustrated how the state-space system can be used for a univariate time series. In the next sections, we also consider trend analyses based on multivariate time series models.

## 4 Trend analysis of the airborne fraction

It follows immediately from Eq. (2) that we can measure the airborne fraction $AF_t$ in two alternative ways:

$$AF_t^{(1)} = \frac{G_t^{ATM}}{E_t^{ANT}}, \qquad AF_t^{(2)} = \frac{E_t^{ANT} - S_t^O - S_t^L}{E_t^{ANT}} = AF_t^{(1)} + \xi_t, \tag{8}$$

where $\xi_t = B_t^{IM}/E_t^{ANT}$, since $E_t^{ANT} - S_t^O - S_t^L = G_t + B_t^{IM}$. Although the two quantities in (8) measure the same underlying object (the airborne fraction $AF_t$), they differ in practice, because of a non-zero budget imbalance, i.e. $\xi_t \neq 0$. Our statistical analysis implies that $\xi_t$ is a well-behaved zero-mean and covariance stationary error process.

We consider our baseline local linear trend model of Sect. 3 for each of the objects, that is,

$$y_t = AF_t^{(i)} = T_t^{(i)} + \epsilon_t^{(i)},$$

for $i = 1, 2$, where the trend $T_t^{(i)}$ is specified in (6) and with error $\epsilon_t^{(i)}$. Table 1 reports the output of the estimation, using the state-space system and the Kalman filter. The first part of Table 1 presents estimates of the standard deviations of the observation error term $\epsilon_t^{(i)}$ and the trend error term $\eta_t^{(i)}$, as well as the estimate of the slope parameter $\beta$, including the estimated standard error (s.e.) of $\hat{\beta}$ and the resulting $t$-statistic, $t$-stat $= \hat{\beta} / \text{s.e.}(\hat{\beta})$. Based on these estimation results, we can formally test hypotheses of the type

$$H_0 : \beta = 0 \quad \text{against} \quad H_1 : \beta \neq 0, \tag{9}$$

or, more relevantly,

$$H_0 : \beta = 0 \quad \text{against} \quad H_1 : \beta > 0. \tag{10}$$

By using the normal approximation to the $t$-distribution and for a $95\%$ confidence level, the critical value for the test (9) is 1.96, and for (10) it is 1.645. In case of the airborne fraction, we are interested in testing (10). It is evident from Table 1 that we cannot reject $H_0$ in this case ($p$-values 0.2711 and 0.4042, respectively). In other words, although the estimate $\hat{\beta}$ is positive, we cannot conclude, statistically at $95\%$ confidence, that the airborne fraction is increasing over time.

Table 1 also contains diagnostic statistics for the standardized prediction residual $u_t$ based on

$$y_t - \mathbb{E}(y_t | y_1, \ldots, y_{t-1}; A, B, \Sigma_\xi, \Sigma_\kappa),$$

for $t = 1, \ldots, n$, and where $\Sigma_\xi$ and $\Sigma_\kappa$ are replaced by their respective maximum likelihood estimates. Under the assumption that the local linear trend model is correctly specified for the time series $y_t$, the residuals $u_t$ are Gaussian iid; see Durbin and Koopman (2012), p. 38. To verify these properties of $u_t$ empirically, we consider two residual diagnostic statistics: the normality test statistic $N$ of Jarque and Bera (1987) and the serial correlation test statistic $DW$ of Durbin and Watson (1971). As a goodness-of-fit statistic, we consider the $R_d^2$ which is a relative measure of model fit against a random walk model. The statistic is defined in a similar way as the standard regression fit measure $R^2$, we have

$$R_d^2 = 1 - \frac{\sum_{t=2}^n u_t^2}{\sum_{t=2}^n [(y_t - y_{t-1}) - m]^2}, \qquad m = (n-1)^{-1} \sum_{t=2}^n (y_t - y_{t-1}).$$

The reported diagnostic statistics and goodness-of-fit in Table 1 are satisfactory for the time series $AF_t^{(1)}$ and $AF_t^{(2)}$. We may conclude from these results that the local linear trend model (6)-(7) provides an adequate description of the dynamic features in the time series. Since the $AF_t^{(2)}$ is well-described within our state-space framework, the extra error term $\xi_t = B_t^{IM} / E_t^{ANT}$ in $AF_t^{(2)}$, as introduced by the budget imbalance term in Eq. (8), is well-behaved. Hence the assumptions underlying the state-space system appear to be valid.

**Table 1.** Univariate analysis of the airborne fraction

| | Parameter estimates | | | | | Diagnostics | | |
|---|---|---|---|---|---|---|---|---|
| | $\widehat{\sigma}_\epsilon$ | $\widehat{\sigma}_\eta$ | $\widehat{\beta}$ | s.e.$(\widehat{\beta})$ | $t$-stat$(\widehat{\beta})$ | $N$ | $R_d^2$ | $DW$ |
| $AF_t^{(1)}$ | 0.1357 | 0.0101 | 0.00109 | 0.00179 | 0.60934 | 0.274 | 0.442 | 1.829 |
| $AF_t^{(2)}$ | 0.1353 | 0.0122 | 0.00049 | 0.00203 | 0.24246 | 2.324 | 0.489 | 1.991 |

We report parameter estimates for the standard deviations $\sigma_\epsilon$ and $\sigma_\eta$, and slope coefficient $\beta$ together with its standard error (s.e.) and $t$-statistic ($t$-stat). We further report the normality ($N$) test, the goodness-of-fit statistic $R_D^2$ and the Durbin-Watson ($DW$) test statistic for serial correlation; all computed for the standardized prediction errors $u_t$ which are obtained from the Kalman filter. The normality test $N$ is the $\chi^2$ distributed, with 2 degrees of freedom, statistic of Jarque and Bera (1987) with its 95% critical value of 5.99; the statistic relies on the sample estimates of skewness and kurtosis of $u_t$. The goodness-of-fit statistic $R_d^2$ is defined as $1 - ESS/DSS$ where $ESS = \sum_{t=2}^n u_t^2$ and $DSS = \sum_{t=2}^n [(y_t - y_{t-1}) - m]^2$ with $m = (n-2)^{-1} \sum_{t=2}^n (y_t - y_{t-1})$. The Durbin-Watson $DW$ test statistic is developed by Durbin and Watson (1971), where also its critical values are tabulated. If $DW = 2$ the sequence $u_t$ is serially uncorrelated; if $DW < 2$ there is evidence that the errors $u_t$ are positively autocorrelated; if $DW > 2$ there is evidence that the errors $u_t$ are negatively autocorrelated.

The state-space system allows both measures for the airborne fraction, $AF_t^{(1)}$ and $AF_t^{(2)}$, to be included in a single model with the purpose to improve the quality of the trend estimation and inference. We begin with an "uninformed" system using two different trend components, $T_t^{(1)}$ and $T_t^{(2)}$, both specified as (6), for the two time series. We have

$$y_t = \begin{bmatrix} AF_t^{(1)} \\ AF_t^{(2)} \end{bmatrix} = \begin{bmatrix} G_t^{ATM}/E_t^{ANT} \\ 1 - (S_t^{OCEAN} + S_t^{LAND})/E_t^{ANT} \end{bmatrix} = \begin{bmatrix} T_t^{(1)} \\ T_t^{(2)} \end{bmatrix} + \begin{bmatrix} \epsilon_t^{(1)} \\ \epsilon_t^{(2)} \end{bmatrix}, \tag{11}$$

where the error terms $\epsilon_t^{(i)}$, for $i = 1, 2$, are correlated and their correlation coefficient can be estimated by the method of maximum likelihood together with the other parameters. The estimation results for this model are presented in Panel A of Table 2. The main difference to Table 1 is the inclusion of the estimated correlation matrix for $(\epsilon_t^{(1)}, \epsilon_t^{(2)})$. The diagnostic test statistics are reasonable. In comparison with the univariate analysis, the goodness-of-fit values for $R_d^2$ are slightly higher for the multivariate model. Hence we trust the model to be a good representation of the data. Furthermore, the slope is estimated to be

positive in both cases (that is $\hat{\beta} > 0$). However, when testing the null hypothesis given in (10), we cannot reject the hypothesis that the slopes are zero ($p$-values 0.3753 and 0.4895, respectively).

Since the two quantities in (8) are measuring the same object, the airborne fraction, we now force the state-space system to recognize that these data are driven by the same underlying common trend, $T_t^A$, but with possibly different error terms $\epsilon_t^{(1)}$ and

$\epsilon_t^{(2)}$. In other words, we consider

$$y_t = \begin{bmatrix} AF_t^{(1)} \\ AF_t^{(2)} \end{bmatrix} = \begin{bmatrix} G_t^{ATM}/E_t^{ANT} \\ 1 - (S_t^{OCEAN} + S_t^{LAND})/E_t^{ANT} \end{bmatrix} = \begin{bmatrix} T_t^A \\ T_t^A \end{bmatrix} + \begin{bmatrix} \epsilon_t^{(1)} \\ \epsilon_t^{(2)} \end{bmatrix}. \tag{12}$$

The output of the estimation of this system is shown in Panel B of Table 2; the estimated common trend and the data are plotted in Fig. 1. A slight deterioration of the diagnostic statistics is to be expected when introducing a common trend into the system,

**Table 2.** Multivariate analysis of the airborne fraction

| | Parameter estimates | | | | | Correlation matrix ($\epsilon$) | | Diagnostics | | |
|---|---|---|---|---|---|---|---|---|---|---|
| **Panel A: Two individual trends as in Eq. (11).** | | | | | | | | | | |
| | $\widehat{\sigma}_\epsilon$ | $\widehat{\sigma}_\eta$ | $\widehat{\beta}$ | s.e.$(\widehat{\beta})$ | $t$-stat$(\widehat{\beta})$ | $AF^{(1)}$ | $AF^{(2)}$ | $N$ | $R_d^2$ | $DW$ |
| $AF^{(1)}$ | 0.1268 | 0.0333 | 0.00146 | 0.00459 | 0.31797 | 1.0000 | 0.7612 | 0.603 | 0.484 | 2.0152 |
| $AF^{(2)}$ | 0.1307 | 0.0274 | 0.00010 | 0.00383 | 0.02629 | 0.7612 | 1.0000 | 1.469 | 0.525 | 2.0853 |
| **Panel B: One common trend as in Eq. (12).** | | | | | | | | | | |
| | $\widehat{\sigma}_\epsilon$ | $\widehat{\sigma}_\eta$ | $\widehat{\beta}$ | s.e.$(\widehat{\beta})$ | $t$-stat$(\widehat{\beta})$ | $AF^{(1)}$ | $AF^{(2)}$ | $N$ | $R_d^2$ | $DW$ |
| $AF^{(1)}$ | 0.1370 | 7.2e-09 | 0.00073 | 0.00095 | 0.77258 | 1.0000 | 0.5518 | 0.245 | 0.470 | 1.8722 |
| $AF^{(2)}$ | 0.1375 | – | – | – | – | 0.5518 | 1.0000 | 2.573 | 0.516 | 1.9820 |

We report parameter estimates for the standard deviations $\sigma_\epsilon^{(i)}$ and $\sigma_\eta^{(i)}$, for $i=1,2$, correlation matrix for $\epsilon_t$, and slope coefficient $\beta$ together with its standard error (s.e.) and $t$-statistic ($t$-stat). We further report the normality ($N$) test, the goodness-of-fit statistic $R_D^2$ and the Durbin-Watson ($DW$) test statistic for serial correlation; for details see Table 1. In Panel B, the trend cofficients ($\sigma_\eta$ and $\beta$) for $AF^{(2)}$ are the same as for $AF^{(1)}$ given the construction of model (12).

but the diagnostic statistics are still such that we can accept (12) as a plausible model. For the estimate of the slope $\widehat{\beta}$, we find a larger $t$-statistic in absolute value than in the uninformed model, indicating that the restriction to the common trend increases the precision of the estimates. An explanation of this finding is that the informed system has used twice as many observations for estimating the trend compared to the uninformed system. The hypothesis test (10) reveals that the estimate of the slope parameter, although again positive, is still not statistically different from zero ($p$-value 0.2199).

**Figure 1.** Estimated trend $T_t^A$ of the airborne fraction from Model (12).

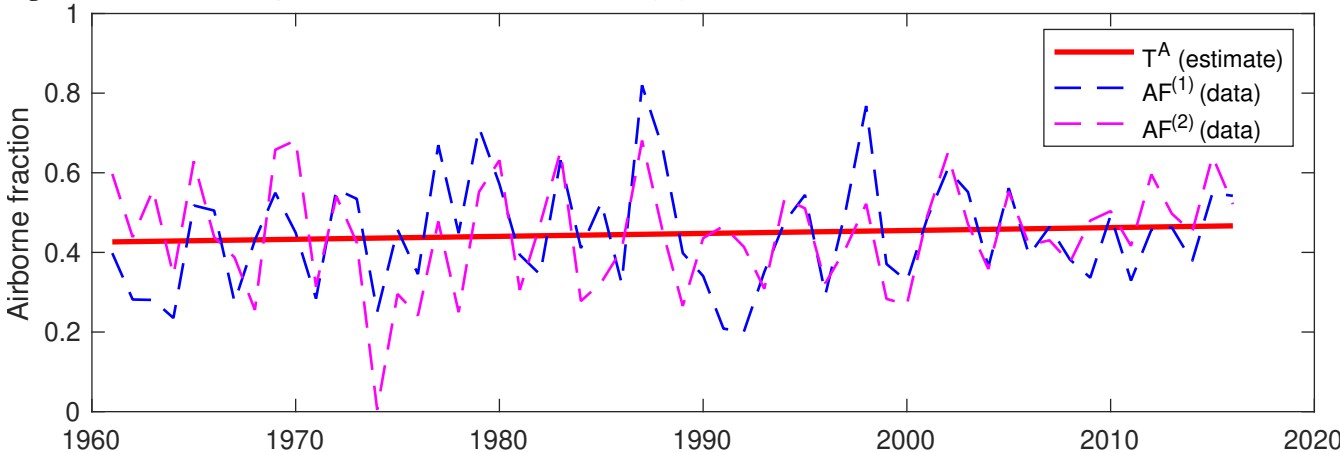

## 5   Trend analysis of the CO$_2$ sink rate

In this section, we analyse the CO$_2$ sink rate in the same way as the airborne fraction above. Analogously we can define two alternative versions of the sink rate:

$$k_{S,t}^{(1)} = \frac{S_t^O + S_t^L}{C_t}, \qquad k_{S,t}^{(2)} = \frac{E_t^{ANT} - G_t}{C_t} = k_{S,t}^{(1)} + \xi_t, \tag{13}$$

where now $\xi_t = B_t^{IM}/C_t$ and where we have used Eq. (2). As was the case for the airborne fraction, these two quantities are measuring the same underlying object (the sink rate, $k_{S,t}$) but differ in practice because of a non-zero budget imbalance, i.e. $\xi_t \neq 0$.

   The basic (univariate) local linear trend model for each of these objects is then given by

$$y_t = k_{S,t}^{(i)} = T_t^{(i)} + \epsilon_t^{(i)},$$

for $i = 1, 2$, where $T_t^{(i)}$ is specified as in (6). When the model is cast in the state-space system, the parameters can be estimated for each of the data series individually. The estimation results are presented in Table 3. The diagnostic statistics are satisfactory, and we conclude again that the error term $\xi_t = B_t^{IM}/C_t$ is well-behaved, in the sense that the assumptions underlying the state-space system appear to be valid, also for the alternative sink rate data, $k_{S,t}^{(2)}$. Even though the estimates of the slopes are negative, we cannot reject the null hypothesis of $\beta = 0$ ($p$-values 0.2233 and 0.0761, respectively). We still consider a one-sided test as

in (10) but now the relevant alternative hypothesis is $H_1 : \beta < 0$.

**Table 3.** Univariate analysis of the CO$_2$ sink rate

| | Parameter estimates | | | | | Diagnostics | | |
|---|---|---|---|---|---|---|---|---|
| | $\widehat{\sigma}_\epsilon$ | $\widehat{\sigma}_\eta$ | $\widehat{\beta}$ | s.e.($\widehat{\beta}$) | $t$-stat($\widehat{\beta}$) | $N$ | $R_d^2$ | $DW$ |
| $k_S^{(1)}$ | 0.0066 | 8.8077e-04 | -0.00010 | 0.00013 | -0.76117 | 4.880 | 0.464 | 1.968 |
| $k_S^{(2)}$ | 0.0063 | 6.3982e-04 | -0.00015 | 0.00010 | -1.43179 | 0.967 | 0.442 | 1.875 |

We report parameter estimates for the standard deviations $\sigma_\epsilon$ and $\sigma_\eta$, and slope coefficient $\beta$ together with its standard error (s.e.) and $t$-statistic ($t$-stat). We further report the normality ($N$) test, the goodness-of-fit statistic $R_D^2$ and the Durbin-Watson ($DW$) test statistic for serial correlation; all computed for the standardized prediction errors $u_t$ which are obtained from the Kalman filter; for details see Table 1.

   Analogously to the airborne fraction above, these data can be put in a joint "uninformed" system with two different trend components, and we have

$$y_t = \begin{bmatrix} k_{S,t}^{(1)} \\ k_{S,t}^{(2)} \end{bmatrix} = \begin{bmatrix} (S_t^O + S_t^L)/C_t \\ (E_t^{ANT} - G_t)/C_t \end{bmatrix} = \begin{bmatrix} T_t^{(1)} \\ T_t^{(2)} \end{bmatrix} + \begin{bmatrix} \epsilon_t^{(1)} \\ \epsilon_t^{(2)} \end{bmatrix}, \tag{14}$$

which can be compared with model (11). The estimation results for this model are reported in Panel A of Table 4. Although the slope estimates are negative, they are not significant ($p$-values 0.3106 and 0.1947, respectively).

**Table 4.** Multivariate analysis of the $CO_2$ sink rate

| | Parameter estimates | | | | | Correlation matrix ($\epsilon$) | | Diagnostics | | |
|---|---|---|---|---|---|---|---|---|---|---|
| Panel A: Two individual trends as in Eq. (14). | | | | | | | | | | |
| | $\widehat{\sigma}_\epsilon$ | $\widehat{\sigma}_\eta$ | $\widehat{\beta}$ | s.e.$(\widehat{\beta})$ | $t$-stat$(\widehat{\beta})$ | $AF^{(1)}$ | $AF^{(2)}$ | $N$ | $R_d^2$ | $DW$ |
| $k_S^{(1)}$ | 0.0064 | 0.0015 | -0.00010 | 0.00020 | -0.49406 | 1.0000 | 0.7733 | 3.348 | 0.511 | 2.0233 |
| $k_S^{(2)}$ | 0.0060 | 0.0014 | -0.00017 | 0.00020 | -0.86071 | 0.7733 | 1.0000 | 1.365 | 0.488 | 2.0185 |
| Panel B: One common trend as in Eq. (15). | | | | | | | | | | |
| | $\widehat{\sigma}_\epsilon$ | $\widehat{\sigma}_\eta$ | $\widehat{\beta}$ | s.e.$(\widehat{\beta})$ | $t$-stat$(\widehat{\beta})$ | $k_S^{(1)}$ | $k_S^{(2)}$ | $N$ | $R_d^2$ | $DW$ |
| $k_S^{(1)}$ | 0.0068 | 4.1762e-09 | -0.00014 | 0.00005 | -2.99145 | 1.0000 | 0.5621 | 4.012 | 0.499 | 2.0276 |
| $k_S^{(2)}$ | 0.0065 | – | – | – | – | 0.5621 | 1.0000 | 0.090 | 0.474 | 1.7967 |

We report parameter estimates for the standard deviations $\sigma_\epsilon^{(i)}$ and $\sigma_\eta^{(i)}$, for $i = 1, 2$, correlation matrix for $\epsilon_t$, and slope coefficient $\beta$ together with its standard error (s.e.) and $t$-statistic ($t$-stat). We further report the normality ($N$) test, the goodness-of-fit statistic $R_D^2$ and the Durbin-Watson ($DW$) test statistic for serial correlation; for details see Table 1. In Panel B, the trend cofficients ($\sigma_\eta$ and $\beta$) for $k_S^{(2)}$ are the same as for $k_S^{(1)}$ given the construction of model (15).

Finally, we consider the state-space system that imposes a common trend for both time series, $T_t^S$, that is

$$y_t = \begin{bmatrix} k_{S,t}^{(1)} \\ k_{S,t}^{(2)} \end{bmatrix} = \begin{bmatrix} (S_t^O + S_t^L)/C_t \\ (E_t^{ANT} - G_t)/C_t \end{bmatrix} = \begin{bmatrix} T_t^S \\ T_t^S \end{bmatrix} + \begin{bmatrix} \epsilon_t^{(1)} \\ \epsilon_t^{(2)} \end{bmatrix}, \tag{15}$$

which can be compared with model (12). The estimation results are presented in Panel B of Table 4. Similar to the analysis

of the airborne fraction in the previous section, the diagnostic statistics are somewhat worse for the less flexible system with a common trend. However, the diagnostics are still satisfactory while the goodness-of-fit statistics have improved overall. The estimate of the slope is

$$\widehat{\beta} = -0.00014,$$

and this estimate is statistically significant: we reject the hypothesis $H_0 : \beta = 0$ in favor of $H_1 : \beta < 0$ at a 95% confidence

level ($p$-value 0.0014). The mean of the sink rate (calculated using either data set $k_S^{(1)}$ or $k_S^{(2)}$) is 0.0258. It follows that we estimate the sink rate to be decreasing with approximately $0.00014/0.0258 = 0.54\%$ every year. The estimated trend and the data are plotted in Fig. 2.

The state-space system is also well-suited for forecasting; see Durbin and Koopman (2012). Using model (15), we forecast the sink rate 25 years ahead in time. The output is presented in Fig. 3. For reference, the forecasts coming from an autoregressive

model of order one (AR1) are also presented. The downward trend in the forecasts from the state-space model is the result of the negative estimate of $\beta$. Under current conditions, the forecast implies that in approximately 15 years, the sink rate will have declined to below 2%. Conversely, the autoregressive model produces forecasts that converge to the mean of the original data series.

It is important to recognise that the validity of these forecasts are conditional on the assumption that the sink rate, $k_S$, is linear in concentrations. As seen from the analysis above, cf. also Fig. 2, this assumption has been approximately satisfied over the time period considered in this paper, but whether it will continue to be accurate is an open question. (Cf. Appendix A for a discussion of the possible future behaviour of the sink rate.) The model-based forecasts of Fig. 3 should be seen in this light: these forecasts are obtained under the assumption that the sink rate will continue to be approximately linear in concentrations. Whether this assumption is reasonable is an interesting question beyond the scope of the present study.

**Figure 2.** Estimated trend $T_t^S$ of the $CO_2$ sink rate from Model (15).

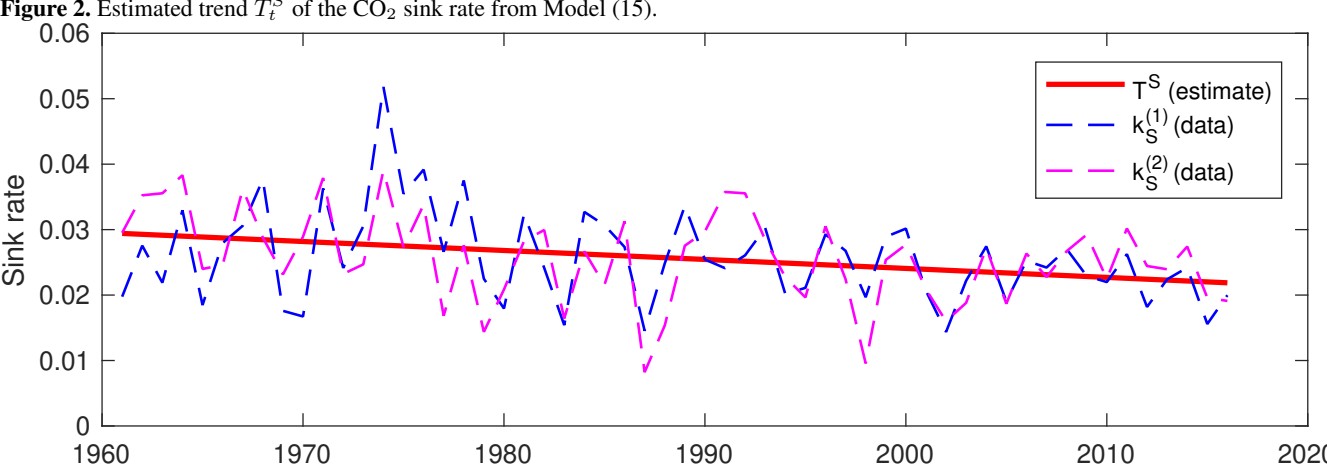

## 6   Trend analysis of the ocean and land sink rates

We may conclude from the analysis in the previous section that the combined (land plus ocean) sink rate appears to be decreasing. To investigate this finding in more detail, we study two alternative models, which consider the two sink variables separately. The first model specifies local linear trends for ocean and land sink rates, i.e.,

$$y_t = \begin{bmatrix} k_{O,t} \\ k_{L,t} \end{bmatrix} = \begin{bmatrix} S_t^O/C_t \\ S_t^L/C_t \end{bmatrix} = \begin{bmatrix} T_t^O \\ T_t^L \end{bmatrix} + \begin{bmatrix} \epsilon_t^{(1)} \\ \epsilon_t^{(2)} \end{bmatrix}, \tag{16}$$

where the time series $k_{O,t}$ and $k_{L,t}$ are defined in (4) while the trend variables $T_t^O$ and $T_t^L$ are specified as in (6). To inform the state-space system of the structure of the carbon budget, we also consider the model equations

$$y_t = \begin{bmatrix} k_{O,t} \\ k_{L,t} \\ k_{S,t} \end{bmatrix} = \begin{bmatrix} S_t^O/C_t \\ S_t^L/C_t \\ (E_t^{ANT} - G_t)/C_t \end{bmatrix} = \begin{bmatrix} T_t^O \\ T_t^L \\ T_t^O + T_t^L \end{bmatrix} + \begin{bmatrix} \epsilon_t^{(1)} \\ \epsilon_t^{(2)} \\ \epsilon_t^{(3)} \end{bmatrix}. \tag{17}$$

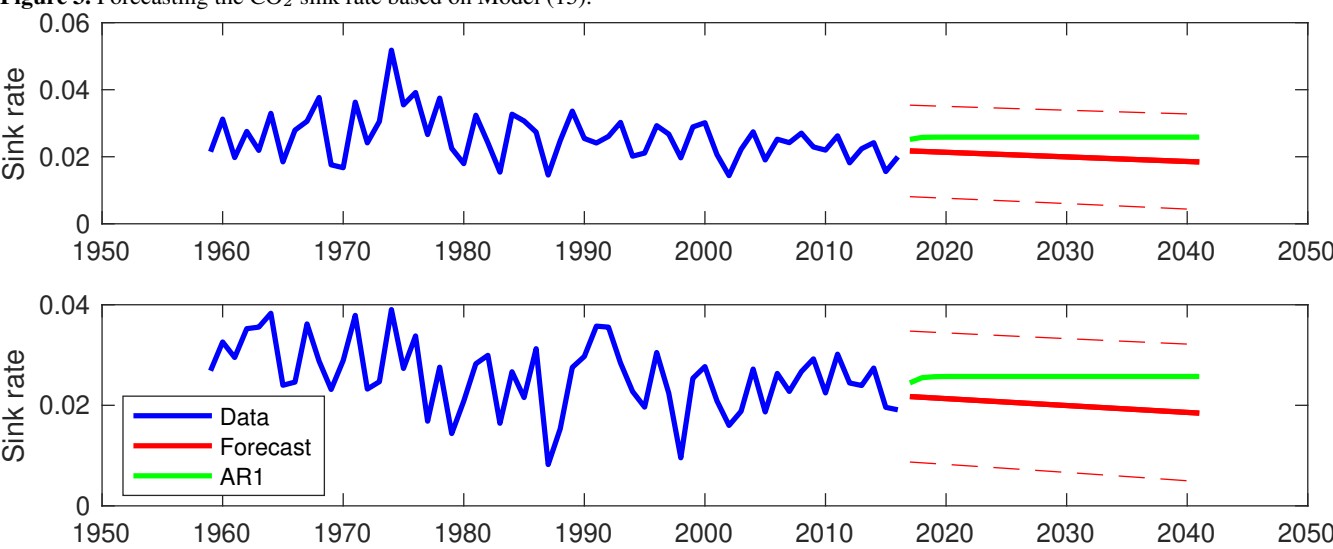

**Figure 3.** Forecasting the $CO_2$ sink rate based on Model (15).

**Figure 3.** The blue solid line represents the data, while the red solid line represents the point forecasts from the Kalman filter with the unknown parameters estimated by maximum likelihood. The dashed red lines are $95\%$ confidence bands ($\pm 1.96$ standard deviation) for the forecasts. The green line represents the forecasts from an autoregressive model of order one.

This trivariate model equation can be cast in the state-space system (5) as well. The model specification has two independent trend processes of the form (6) for land and ocean sinks. Since $k_{S,t} = k_{O,t} + k_{L,t}$, the time series $k_{S,t}$ of combined ocean and land sinks must feature the sum of the two trend processes for the individual sinks as its trend process.

The estimation results for these two model specifications are presented in Table 5. The residual diagnostic statistics $N$ and $DW$ are satisfactory, but we are particularly interested in the estimates of the slope parameters. It seems that most of the decrease in the sink rate can be attributed to the land sink. The slope estimates of the trend driving the ocean sink rate are very close to zero and not statistically significant ($p$-values $0.5227$ and $0.5168$, respectively). On the other hand, the slope estimates of the trend driving the land sink rate are negative for both specifications. In the first model (16), we can reject the hypothesis that the slope of the trend driving the land sink rate is zero, in favor of the one-sided alternative $H_1 : \beta < 0$ at a $95\%$ confidence level ($p$-value of $0.0420$). For the more informed model specification (17), the estimation results are reported in Panel B of Table 5. Here we can reject $H_0$ at a $90\%$ confidence level ($p$-value of $0.0882$). Further, the results show that the estimate of the slope parameter from the land sink rate is equal to the estimate of the slope parameter from the combined sink rate as in Sect. 5, that is, $\hat{\beta} = -0.00014$. In other words, it appears that the decrease in the combined sink rate studied in the previous section is entirely explained by the decrease in the land sink rate.

**Table 5.** Analysis of ocean and land sink rates

Panel A: Two trends, two observation series as in Eq. (16).

| | Parameter estimates | | | | | Correlation matrix ($\epsilon$) | | Diagnostics | | |
| --- | --- | --- | --- | --- | --- | --- | --- | --- | --- | --- |
| | $\widehat{\sigma}_\epsilon$ | $\widehat{\sigma}_\eta$ | $\widehat{\beta}$ | s.e.($\widehat{\beta}$) | $t$-stat($\widehat{\beta}$) | $k_{O,t}$ | $k_{L,t}$ | $N$ | $R_d^2$ | $DW$ |
| $k_{O,t}$ | 0.0001 | 0.00081 | 0.00001 | 0.00011 | 0.057 | 1.00 | -1.00 | 4.839 | 0.0343 | 1.847 |
| $k_{L,t}$ | 0.0067 | 0.00015 | -0.00010 | 0.00006 | -1.728 | -1.00 | 1.00 | 5.332 | 0.513 | 1.908 |

Panel B: Two trends, three observation series as in Eq. (17).

| | $\widehat{\sigma}_\epsilon$ | $\widehat{\sigma}_\eta$ | $\widehat{\beta}$ | s.e.($\widehat{\beta}$) | $t$-stat($\widehat{\beta}$) | $k_{O,t}$ | $k_{L,t}$ | $k_{S,t}$ | $N$ | $R_d^2$ | $DW$ |
| --- | --- | --- | --- | --- | --- | --- | --- | --- | --- | --- | --- |
| $k_{O,t}$ | 0.0001 | 0.00081 | 0.00000 | 0.0001 | 0.0422 | 1.00 | -0.122 | -0.884 | 4.839 | 0.0343 | 1.916 |
| $k_{L,t}$ | 0.0068 | 0.00068 | -0.00014 | 0.0001 | -1.352 | -0.122 | 1.00 | 0.572 | 4.054 | 0.494 | 1.989 |
| $k_{S,t}$ | 0.0065 | - | - | - | - | -0.884 | 0.572 | 1.00 | 1.114 | 0.477 | 1.801 |

We report parameter estimates for standard deviations $\sigma_\epsilon^{(i)}$ and $\sigma_\eta^{(i)}$, for $i = 1, 2, 3$, correlation matrix for $\epsilon_t$, and slope coefficient $\beta$ together with its standard error (s.e.) and $t$-statistic ($t$-stat). We further report the normality ($N$) test, the goodness-of-fit statistic $R_D^2$ and the Durbin-Watson ($DW$) test statistic for serial correlation; for details see Table 1. In Panel B, we have two trends and two sets of trend cofficients ($\sigma_\eta$ and $\beta$) for $k_{O,t}$ and $k_{L,t}$, the trend for $k_{S,t}$ is a combination of the two, given the construction of model (17).

## 7 Discussion

Previous studies of the airborne fraction and the $CO_2$ sink rate have focused on detecting a single linear and deterministic trend in the data of the form $a_0 + a_1 t$, where $a_0, a_1$ are constants (Canadell et al., 2007a; Le Quéré et al., 2009; Knorr, 2009; Raupach et al., 2008, 2014). However, possible statistical difficulties in such analyses have been pointed out in Knorr
(2009). For instance, a linear regression analysis of two or more non-stationary variables can yield invalid inference (Granger and Newbold, 1974). The approach of this paper is to consider the data in a state-space system. In this way, non-stationary components are explicitly modelled as unobserved trend components and inference is valid (e.g., Durbin and Koopman, 2012). Furthermore, the trend specification of the state-space system allows for both deterministic and stochastic trend components.

In some of the "un-informed" models, cf. Table 1, Panel A of Table 2, and Panel A of Table 4, we estimate $\hat{\sigma}_{Slp} > 0$ and,
thus, in these cases, we find evidence of the trend component varying in time. However, in our "informed" models with a single trend object for two alternative time series, the extracted trends are practically deterministic, that is, the estimates of $\sigma_{Slp}$ in Panel B of Tables 2 and 4 are near zero, cf. also Fig. 1 and 2. In conclusion, there is evidence that a simple deterministic trend fits both the airborne fraction and the sink rate data well, although this only becomes evident when incorporating two data sets for each of these objects.

Several studies have highlighted the need for accounting for noise in measurements of climate-related data (Knorr, 2009; Ballantyne et al., 2015). The state-space approach explicitly incorporates such noise in the framework as well. Ballantyne et al. (2015) argue that errors in $E_t^{ANT}$ might be autocorrelated. As shown in Tables 1 through 5, the diagnostic statistics do not indicate that autocorrelated errors pose a serious problem in our specifications. Nevertheless, the state-space framework can incorporate autocorrelated errors in the measurement equation.

Why do we find statistical evidence of a decreasing $CO_2$ sink rate but no evidence of an increasing airborne fraction when these two quantities are closely linked and the data entering the analyses are the same? It was noted in Gloor et al. (2010) that the airborne fraction and the sink rate are actually not as closely linked as they may seem *prima facie*. In particular, an increasing airborne fraction does not necessarily imply a decreasing sink rate and vice versa (Gloor et al., 2010, Section 8).

The findings of this paper supports this claim by providing statistical evidence for a constant airborne fraction but at the same time a decreasing sink rate. Secondly, the concept of an airborne fraction is that of a long-term quantity: the airborne fraction should represent the amount of anthropogenically released $CO_2$ that *eventually* stays in the atmosphere, *after* other fluxes have been taken into account. However, the ratio of the concurrent fluxes, i.e., $G_t/E_t^{ANT}$, is likely a very noisy measurement of this object. Also, as we saw above, it is reasonable to consider sink fluxes, and therefore indirectly $G_t$, as dependent on the

*level* of $CO_2$ in the atmosphere (i.e., $C_t = \sum G_t$), which is not captured by the concurrent ratio $G_t/E_t^{ANT}$. When studying the airborne fraction, it would perhaps be more reasonable to study an object taking this cumulative nature into account, e.g. $\sum G_t / \sum E_t^{ANT} = C_t / \sum E_t^{ANT}$ (in fact, such specifications were often considered in earlier parts of the literature, e.g. Keeling, 1973; Bacastow and Keeling, 1979; Oeschger and Heimann, 1983; Enting and Pearman, 1986). However, cumulative statistics of this type would present other difficulties. The dominance of the long-term history may mask sudden changes, for

example. In contrast, the sink rate $S_t/C_t$, as a flow-to-stock ratio, is immediately compatible with the underlying theory, at least as long as the linear approximation of the relationship between $S_t$ and $C_t$, such as was made in e.g. Gloor et al. (2010) and Rayner et al. (2015), is adequate.

    What are possible physical explanations for the apparent decrease in the sink rate? Raupach (2013) argues that a necessary condition for a constant sink rate is that the so-called "LinExp" assumption holds, i.e., that the sink fluxes $S_t^O$ and $S_t^L$ are

linear in concentrations $C_t$ ("Lin") and that emissions ($E_t^{ANT}$) grow exponentially ("Exp"). Constancy of the airborne fraction rests on a similar "LinExp" argument. Since we find no statistical evidence that the airborne fraction, $AF_t$, and the ocean sink rate, $k_{O,t}$, are non-constant in time, it is unlikely that the "Exp" assumption is grossly violated over the observation period considered in this paper. In contrast, it was found above that the efficiency of the land sink, $k_{L,t}$, is decreasing. A plausible explanation of these findings is that the "Lin" assumption no longer holds for the land sink, for instance because the terrestrial

sink could be slowly saturating (Canadell et al., 2007b). In Appendix A we give a formal argument for how this could lead to the findings documented above. In particular, we show that the findings of this paper can be explained by the land sink's response to high atmospheric $CO_2$ concentrations: it is plausible that due to a rising level of $CO_2$ concentration, non-linear effects in the terrestrial $CO_2$ carbon cycle have become noticeable. If this is indeed the case, it has obvious consequences for our understanding of the carbon cycle and should be a cause for substantial concern (Gloor et al., 2010, p. 7740). However,

although this explanation of our findings is consistent with the data, we can not conclude that it is the only possible explanation. Further research into the underlying reasons for the decreasing sink rate would be very valuable and are left for future work.

    It is possible that the analyses conducted above are influenced by external natural events such as ENSO, volcanic eruptions, and the like (Frölicher et al., 2013). The state-space system used in this paper can explicitly account for such effects through the additive error terms $\epsilon_t$, cf. Eq. (5). To verify that the approach is indeed robust to such external and transitory events,

we have also conducted our analyses using 5-year average data. The findings from the estimated state-space system for these

time series of averages confirm those reported above: In the joint estimation, we find no statistical evidence of a trend in the airborne fraction ($p$-value of $0.3214$), and we do find statistical evidence of a decreasing trend in the sink rate ($p$-value of $0.00064$). We conclude that the findings of this paper are not likely to be driven by external natural events such as ENSO and volcanic eruptions. We also considered 2-, 3-, and 4-year averages with similar results. We present details of this analysis in

the Supplementary Material file (Sect. 3). To further check the robustness of the results, we examined whether there are any of the observations in the data set which are particularly influential. Statistically influential observations could be due to outliers, caused for instance by external natural events, such as the ones mentioned above. Using a statistic called Cook's distance (Cook, 1977, 1979; Atkinson et al., 1997), which is a measure of how influential a given observation is on the analysis, we did not find evidence of any one observation being particularly influential. Similarly, we tried estimating the slope parameter

$\beta$ after deleting the $t$'th observation for each time point $t$ in the sample, i.e., for $t = 1959, 1960, \ldots, 2016$; the estimates of the slope parameter found in this way were very stable, which is further evidence of the robustness of the analyses to potential outliers and external events. Details can be found in the Supplementary Material file (Sect. 2.1).

This paper considers data recorded at a yearly frequency, while many of the previous studies of the airborne fraction and the sink rate use monthly data. The advantage of using monthly data is obvious: more observations. However, there are also some

disadvantages. For instance, while the $CO_2$ concentration $C_t$ (and therefore also the growth rate $G_t$) are recorded every month, these data contain a strong seasonal component induced by the photosynthesis/respiration cycle of terrestrial vegetation. This seasonality needs to be accounted for in some way; for instance, Raupach et al. (2014) smooth the data using a 15-month running mean. In contrast, some of the other data are recorded only yearly; for instance, emissions data available to us, $E_t^{ANT}$. In this case, Raupach et al. (2014) use linear interpolation to get monthly estimates of emissions. Such transformations of the

data, i.e., smoothing or interpolation, might introduce new and complicated errors, possibly invalidating the analyses. For these reasons, we prefer to work with yearly data.

## 8   Conclusions

We have argued that the state-space system can be a useful approach to analyze possible trends in the airborne fraction of anthropogenically released $CO_2$ and in the $CO_2$ sink rate. We have shown that deterministic and stochastic trend processes can

be explicitly and jointly incorporated as unobserved components, allowing for valid inference, even when the observed time series are non-stationary. The state-space framework also allows for the incorporation of multiple data sets for the same object, which increases reliability of the resulting estimates.

We estimate a positive, yet statistically insignificant, slope in the data for the airborne fraction. Using two alternative time series for the sink rate and imposing a common trend, we obtain a significantly negative deterministic trend. Our analyses

support the conclusions as set out by Raupach et al. (2014): the rate at which the combined (ocean plus land) sink takes up $CO_2$ from the atmosphere seems to be decreasing. The best estimate resulting from our state-space system is that the $CO_2$ sink rate, and therefore the efficiency with which the combined land and ocean sink is absorbing carbon from the atmosphere, is decreasing by $0.54\%$ per year. We do not find evidence of this rate itself changing over time.

Finally, there is tentative evidence that the decrease in the sink rate is mainly driven by a weakening uptake in the land sink. This could be the result of non-linearities in the response of the terrestrial carbon sink to the level of atmospheric concentrations of $CO_2$. That is, although the land sink is itself increasing and thus continuing to take up a large part of anthropogenically emitted $CO_2$, as also noted recently by e.g. Rayner et al. (2015), Keenan et al. (2016), and Fernández-Martínez et al. (2019), the *rate* of this uptake appears to be decreasing. The statistical evidence for this is not strong, however, and we suggest that additional research must be conducted to further investigate this question.

*Data availability.*  The data used in this paper are available at the website of the Global Carbon Project (https://www.globalcarbonproject.org).

*Author contributions.*  All authors contributed equally to the paper.

*Competing interests.*  No competing interests are present.

*Acknowledgements.*  We would like to thank Corinne Le Quéré for permission to use the data set of Le Quéré et al. (2018), as well as for useful comments on the manuscript. We would also like to thank two anonymous referees and the associate editor for constructive and helpful comments. MB and EH acknowledge financial support from the Independent Research Fund Denmark for the project Econometric Modeling of Climate Change.

## Appendix A: Linear approximation of the relation of land sink and concentrations

In this appendix, we argue that the levels of atmospheric concentrations of $CO_2$ may have risen to a point where a linear expansion of the logarithmic Bacastow and Keeling (1973) formula, describing the flux of $CO_2$ into the land sink, is no longer sufficient. Consequently, the "Lin" assumption of Raupach (2013) might be violated for the land sink, implying that second-order effects may be driving the negative slope of the sink rate that we document in this paper.

From Eq. (4) we obtain the relation

$$S_t^L = k_{L,t} \cdot C_t,$$

which implies that the flux of $CO_2$ to the land sink is linear in $C_t$, where $k_{L,t}$ would then be treated as a constant. On the other hand, a decreasing $k_{L,t}$ implies that the efficiency with which the land sink absorbs $CO_2$ is decreasing, i.e., that the flux of $CO_2$ to the land sink is non-linear in $C_t$ and this non-linearity is such that the efficiency is decreasing. These statements are consistent with simulation results from climate cycle models (Friedlingstein et al., 2006). Here we illustrate mathematically how such non-linearities can arise.

The precise relationship between $S_t^L$ and $C_t$ still alludes us but Bacastow and Keeling (1973), p. 94, suggest that (in our notation):

$$S_t^L \approx \alpha \log(1 + C_t/\mathcal{C}^0),$$

where $\alpha$ is a constant and $\mathcal{C}^0 = 591.30$ GtC is the amount of $CO_2$ in the atmosphere in pre-industrial times. Using this function, we can write a second-order Taylor expansion

$$S_t^L \approx \alpha \log(1 + C_t/\mathcal{C}^0) \approx \alpha \frac{C_t}{\mathcal{C}^0} - \frac{1}{2}\alpha \left(\frac{C_t}{\mathcal{C}^0}\right)^2.$$

Thus, if $\mathcal{C}^0$ is large compared to $C_t$, this implies that the squared term in the above equation is small and thus that a linear specification between $S_t^L$ and $C_t$ is reasonable. However, once $C_t$ becomes large compared to $\mathcal{C}^0$, this shows that the estimated sink rate will be found to be decreasing. To see this, use the Taylor expansion to write

$$S_t^L \approx k_{L,t} C_t,$$

where

$$k_{L,t} = \frac{\alpha}{\mathcal{C}^0} - \frac{1}{2}\frac{\alpha}{\mathcal{C}^0}\frac{C_t}{\mathcal{C}^0}$$

is decreasing in $C_t$. In our data, we have $C_{1959} \approx 80$ GtC and $C_{2016} \approx 267$ GtC, resulting in $C_{1959}/\mathcal{C}^0 \approx 14\%$ and $C_{2016}/\mathcal{C}^0 \approx 45\%$. In other words, the linear approximation to the Bacastow and Keeling model of the land sink flux might have been reasonable in the past, since $C_{1959}/\mathcal{C}^0 \approx 14\%$, but is likely misspecified in the present, since $C_{2016}/\mathcal{C}^0 \approx 45\%$. That is, if this model is accurate, then a decreasing (land) sink rate indicates that we have entered a regime of atmospheric $CO_2$ concentrations, where the linear approximation breaks down and higher order terms should be taken into account.

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
