# Peer review of "Trend analysis of the airborne fraction and sink rate of anthropogenically released $CO_2$"

_Biogeosciences, 2018_

## Referee Comment (RC1) · Anonymous Referee #1 · 4 Jan 2019

In the manuscript 'Trend analysis of the airborne fraction and sink rate of anthropogenically released CO2 ' by Bennedsen and coauthors investigate the long-term trends in the CO2 airborne fraction and the combined land and ocean sink rate of CO2 using a state space system that allows to compute local linear trends and hence to account for non-stationnarity in the variance.

I much appreciate this work which is not only highly relevant in the context of understanding impact of climate change on carbon cycle but also by the use a new statistical approach which is relevant to investigate complex systems such as the Earth system. However, I think this paper needs some clarification that have to be addressed first, and

which prevent me of accepting this paper in its present form. Therefore, I recommend acceptance of this manuscript after some major revisions.

Besides, I don't feel that Biogeosciences is a good journal to host this work given the high technicality of the study. My concerns could be addressed by rewriting the article for a broader audience and further discuss the processes at play.

General Major Comments:

First, this study comes after almost a decade-long research (Knorr (2009), Gloor et al. (2010), and Ballantyne et al. (2015)) on the detection of the changes in AF or sink efficiency and does not provide new findings (e.g., results are in the line of Raupach 2014). Yet this work merits to be acknowledged because it is the first to my knowledge to investigate this long debate on the stationnarity of the AF or SF variations. Here the authors confirm that there is no non-stationnarity in AF and SF using GCP2018 data (from 1959 to 2017). Therefore, I am wondering if it is not the real outcomes of the study ? I mean once the stationarity of the variance is proved, the state space system loses some interest. The potential caveats as suggested by Gloor et al 2010 are removed and thus a simple linear model can be used to estimate trends in AF and SF. Standard statistics can be then used to detect if the signal (the trends) is larger than the noise (the variability).

The second major comment concerns the attribution of the decreasing sink to the land carbon sink. Regarding the shape of the land C sink, we may be interested to test since how many years the land sink has started to decrease. To further this comment, I think that several test of the length of the data and the influence of the sampling are missing in the manuscript. We need to see how far this approach is robust when using, for example, 5-year average data (removing ENSO and volcanoes influence).

My last major comment relates to the use of the "balanced" C budget whereas Le Quéré et al. 2018 provides the Bim terms that could be used as a third entry in you model. I mean does the variance of Bim is steady in time or does it vary ? How far

this terms correlates with AF and SF ? Do you fin a trends in Bim that could explain why the sink rate declines whereas the AF does ? I think all these discussions might consolidate the study.

Specific comments: P1 L4 what do you mean by "balanced carbon budget" ? P1 L4 please clarify this sentence. It is unclear to me what object are you talking about P1 L6 please explain a bit further because a decrease in the sink should end up ultimately by a change in the AF P1 L13 please add the reference period over whch this % are estimated + the reference publication P1 L18 you could acknowledge more recent studies here P2 L5 anthropic = anthropogenic P2 L7 you can remove "which we argue is well designed for the problem at hand Âż P3 L12-16 I think paragraph should be move above and better explain why you are working on the "balanced" hypothesis. The Bim remains small compared to the other terms for example ? P4 L2 could you further explain the meaning of "Using a simplifying linear specification Âż ? P6 L12-15 what about for a lower confidence threshold e.g., 90 % do you get a better agreement ? why such a different in Beta estimates (one order of magnitude) ? P7 L14 please give the estimate of TtA ? besides I think there is a error in Eq 13 with the random noise epsilon. I read it as independent of time. P10 L9-10 the last sentence requires further explanations Figure 3 I don't know what these two panels show. They show the two metrics, correct ? Why giving the confidence interval for 1 sigma whereas most of the statistical test were conducted with a 95 % confidence threshold ? P12 L15 this looks like trivial. I guess that a simple correlation between the SF and LF should lead to the same conclusion. . .

---

## Referee Comment (RC2) · Anonymous Referee #2 · 10 Jan 2019

Bennedsen et al. analyse trends of airborne fraction and sink rates of the global carbon cycle using a Kalman filter approach which should not be very sensitive to variations like El Nino. The analysis is very clearly presented and the results seem robust. Thus in my view the paper should be accepted.

I have very few suggestions - and also a general comment - which does not affect acceptance of the paper though. Firstly in the introduction the authors state: 'a key question is whether the airborne fraction is increasing ' but they do not say why. It would be good if they would add why this is so.

Sentence just above - 24% and 31% - I would add a reference here - and possibly

uncertainties - just for completeness.

When applying the Kalman filter the authors will need to initialize it. I may have missed it but if not it would be good if the authors would add this in the main text.

Finally my comment - while all the results are sound - what the paper does not explain is the true reason for the decrease in sink rates - and thus it is not clear whether a decreasing sink rate is alarming or not. It would be nice if the authors could comment on that - but it is not a necessary condition.

Some earlier papers actually give a clue what the real reason may be.

---

## Author Comment (AC1) · 30 Jan 2019

**Trend analysis of the airborne fraction and sink rate of anthropogenically released $CO_2$**

**Reply to referee report number 1**

Mikkel Bennedsen[1,3], Eric Hillebrand[1,3], and Siem Jan Koopman[2,3]

[1]Department of Economics and Business Economics, Aarhus University, Fuglesangs Allé, 4 8210 Aarhus V, Denmark
[2]Department of Econometrics, School of Business and Economics, Vrije Universiteit Amsterdam, De Boelelaan 1105, 1081 HV Amsterdam, The Netherlands.
[3]Center for Research in Econometric Analysis of Time Series (CREATES), Aarhus University, Fuglesangs Allé, 4 8210 Aarhus V, Denmark

**Correspondence:** Mikkel Bennedsen (mbennedsen@econ.au.dk)

**1   Introduction**

We thank the referee for the insightful comments and the supportive review. We will revise the paper in response to your comments, and we think that the paper will improve substantially as a result. We have also prepared a Supplementary Material file with results of further analyses. We elaborate further in our answers below.

**1.1   Comment 1**

**1.1.1   Referee comment**

*First, this study comes after almost a decade-long research (Knorr (2009), Gloor et al. (2010), and Ballantyne et al. (2015)) on the detection of the changes in AF or sink efficiency and does not provide new findings (e.g., results are in the line of Raupach 2014). Yet this work merits to be acknowledged because it is the first to my knowledge to investigate this long debate on the stationarity of the AF or SF variations. Here the authors confirm that there is no non-stationnarity in AF and SF using GCP2018 data (from 1959 to 2017). Therefore, I am wondering if it is not the real outcomes of the study ? I mean once the stationarity of the variance is proved, the state space system loses some interest. The potential caveats as suggested by Gloor et al 2010 are removed and thus a simple linear model can be used to estimate trends in AF and SF. Standard statistics can be then used to detect if the signal (the trends) is larger than the noise (the variability).*

**1.1.2   Answer**

Thank you for raising these important points. In our approach, stationarity or non-stationarity is a *finding* rather than an *assumption*. We agree with the referee that stationarity of the AF and negative linear trending in the sink rate are the main

results, *ex post*. However, *a priori*, we have formulated a statistical dynamic model that allows for both, non-stationary and stationary processes for $y_t$ and for linear as well as stochastic trending behavior. In the paper, below equation (7), we show that the solution to the difference equation (7) leads to a deterministic time trend when the iid Gaussian random variable $\eta_t$ is zero (effectively when its variance is zero, that is $\sigma_\eta^2 = 0$). In this case, the time series is *trend-stationary* and the dynamic process for $y_t$ does not exhibit a unit-root (which is the case for a *difference-stationary* time series). Since $\sigma_\eta^2$ is estimated using the observations for $y_t$, we can only conclude *ex post* that the time series is trend-stationary. Without the estimation of our state-space model, we could not have arrived at this conclusion.

Our main findings can be summarized as follows. (a) We find no statistical evidence of an increasing airborne fraction, while we do find statistical evidence for a decreasing sink rate. (b) While the findings (a) have also been reported elsewhere, most notably in Raupach et al. (2014), our statistical model does not make any a priori assumptions regarding the stationarity or non-stationary of the series and regarding deterministic or stochastic behaviour of the trends. These findings are thus *results*, as opposed to assumptions, of our approach. Furthermore, we find that we need to estimate our model on *all* the data from the global carbon budget jointly to reach the findings (a).

In the joint estimation, we take all data into account, that is the time series for AF and SR as defined on page 3 of the paper, but also the additional data obtained by assuming that the carbon budget is balanced, which we explain on page 4 of the paper. (Note that we follow the sink rate definition of Raupach (2013) and Raupach et al. (2014), with concentrations in the denominator, not emissions. The sink rate (SR) in our paper is thus not the complement of the airborne fraction.) More specifically, when analyzing the sink fraction, for example, we can opt for either one or both of the two time series:

$$k_S^{(1)} = \frac{S_t^O + S_t^L}{C_t},$$

$$k_S^{(2)} = \frac{E_t - G_t}{C_t},$$

where we exploit the carbon budget equation to obtain $k_S^{(2)}$:

$$E_t = G_t + S_t^O + S_t^L + B_t^{IM}.$$

The budget imbalance $B_t^{IM}$ is a zero-mean noise sequence that represents the measurement errors in the other variables of the carbon budget (Le Quéré et al., 2018). In Table 3 of the main paper we report the results when we estimate our state-space model on $k_S^{(1)}$ and $k_S^{(2)}$ separately. In Table 4 and Figure 2 of the main paper we report the results when we estimate the state-space model on $k_S^{(1)}$ and $k_S^{(2)}$ jointly. It is a feature of the state-space model that it allows for alternative measurements for the same object of interest.

In the Supplemental Material file, we have included Figure 1 that presents the extracted the latent trends, $T_t$ in equation (7), from the separate analysis, and Figure 2 that presents the extracted comment trend $T_t$ from the joint analysis. (This is a replication of Figure 2 in the main paper.) The extraction method is based on the filtering and smoothing approach as discussed in the main paper. The various extracted trends illustrate our points as argued above: in the separate analysis, we obtain (slightly) time-varying stochastic trends, whereas in the joint analysis, we obtain a deterministic linear trend with a significantly negative

slope (compare Panel B of Table 4 in the main paper). We have found similar results for the AF series, which we include in the Supplemental Material file.

**Figure 1.** Univariate estimation of time-varying trend for sink rate, $k_S$.

[Figure]

[Figure]

**Figure 2.** Multivariate estimation of time-varying trend for sink rate, $k_S$.

[Figure]

**1.1.3 Plan to change paper in response**

We will paraphrase the explanations of the AF and SR definitions in Section 2 to show the contribution of the state-space model in the context more crisply.

We will include Figure 1 and the corresponding results for the AF in the Supplementary Material.

**1.2 Comment 2**

**1.2.1 Referee comment**

*The second major comment concerns the attribution of the decreasing sink to the land carbon sink. Regarding the shape of the land C sink, we may be interested to test since how many years the land sink has started to decrease. To further this comment, I think that several test of the length of the data and the influence of the sampling are missing in the manuscript. We need to see how far this approach is robust when using, for example, 5-year average data (removing ENSO and volcanoes influence).*

**1.2.2 Answer**

The referee raises two interesting questions; we treat them one-by-one.

(i) With respect to the land sink rate, we are treating it as a fraction: the ratio of flux in land sink over $CO_2$ concentration in the atmosphere. The land sink flux itself, $S_t^L$, is increasing over time but the land sink rate, $k_{L,t} = S_t^L / C_t$, shows evidence of a decreasing trend. Our paper shows (Figure 2 in this reply) that if we sum up ocean and land sink and use both time series for this object, $k_S^{(1)}$ and $k_S^{(2)}$, jointly, we obtain a significantly negatively sloping deterministic trend. We can of course also consider $k_S^{(1)}$ and $k_S^{(2)}$ separately. The result is shown in Figure 1 of this reply. From the left panel of this figure, one might argue that the negative trend started in the mid 1970s. The right panel, which shows $k_S^{(2)}$, however, does not display such a kink. Finally, we can consider the land sink rate, $k_L$, individually. The result is shown in Figure 3 of this reply. We obtain a deterministic trend with an insignificant negative slope, cf. Table 5 of the main paper.

(ii) We have estimated the state-space model on 5-year average data in order to reduce the impact of effects such as ENSO, volcanic eruptions, and the like. The state-space model estimated on annual data is also capable of accounting for these effects, since it treats them as additive noise in the measurement equation. Repeating the analysis based on 5-year average data, however, provides a way to verify our estimation results and conclusions. (We also considered 2-, 3-, and 4-year averages, with similar results.)

We calculate 5-year non-overlapping averages in order to avoid introducing serial correlation into the time series. Running (i.e., overlapping) averages would necessitate specifying a model to capture this serial correlation, and we think this is a relatively bigger disadvantage than the reduction in the sample size that we incur from non-overlapping averages. Since we have 58 years of data, we calculate an average of the first three years followed by 5-year averages, resulting in 12 observations. The findings from estimating the state-space model on these time series of averages confirm those reported in the main paper: In the joint estimation, we find no statistical evidence of a trend in the airborne fraction (with a $p$-value of $0.32138$), and we do

find statistical evidence of a decreasing trend in the sink rate (with a $p$-value of $0.00064$). Of course, the residual diagnostics for these short time series are not as convincing as those presented in the main paper. The extracted trends from these joint analyses are presented in Figures 4 (airborne fraction) and 5 (sink rate) in this reply. Incidentally, some analyses in earlier studies were based on running averages; we discuss these briefly in the Discussion section of the main paper (P13 L10).

5      We emphasize two points in this context: (1) The state-space model is advantageous in this exercise, since it allows to incorporate the alternative time series for both, AF and SR, which is particularly useful when the sample period is short. (2) The main finding from annual data prevails: In the separate analyses, the trends are estimated as stochastic. Only in the joint analysis do we obtain a deterministic trend and statistical significance for the sink rate.

**Figure 3.** Univariate estimation of time-varying trend for land sink rate, $k_L$.

[Figure]

**1.2.3   Plan to change paper in response**

10      – (i) We will include the separate analysis of the land sink rate ($k_L$) in the Supplementary Material. For completeness, we also submit the ocean sink rate ($k_O$) to the same analysis in the Supplementary Material.

   – (ii) We will include the estimation results based on 5-year averages in the Supplementary Material and briefly discuss these findings in the main paper as well, with a reference to the Supplementary Material for further details. We will emphasize that the findings of the paper are robust to averaging of the data.

**Figure 4.** Multivariate estimation of time-varying trend for AF. Data: 5-year averages (no overlap)

[Figure]

**Figure 5.** Multivariate estimation of time-varying trend for sink rate, $k_S$. Data: 5-year averages (no overlap)

[Figure]

**1.3 Comment 3**

**1.3.1 Referee comment**

*My last major comment relates to the use of the "balanced" C budget whereas Le Quere et al. 2018 provides the Bim terms that could be used as a third entry in you model. I mean does the variance of Bim is steady in time or does it vary ? How far this terms correlates with AF and SF ? Do you fin a trends in Bim that could explain why the sink rate declines whereas the AF does ? I think all these discussions might consolidate the study.*

**1.3.2 Answer**

Thank you for raising this point. By assuming that the carbon budget is balanced, we already include the $B_t^{IM}$ data in the analysis. Specifically, the data on the budget imbalance enters as follows. For the case of the sink rate, the two time series employed are:

$$k_S^{(1)} = \frac{S_t^O + S_t^L}{C_t},$$
$$k_S^{(2)} = \frac{E_t - G_t}{C_t}.$$

Given the carbon budget equation, the latter expression can be written as

$$k_S^{(2)} = \frac{E_t - G_t}{C_t} = \frac{S_t^O + S_t^L + B_t^{IM}}{C_t} = k_S^{(1)} + \xi_t,$$

where

$$\xi_t = \frac{B_t^{IM}}{C_t},$$

can be regarded as an error term. This is the motivation for using the two time series $k_S^{(1)}$ and $k_S^{(2)}$ as data for the same underlying quantity, that is, the sink rate.

Figure 6 plots the time series of $\xi_t$ (left plot) and $B_t^{IM}$ (right plot). Both have a mean that is not significantly different from zero and follow stationary dynamics, albeit with some serial correlation.

In the joint state-space model, both $k_S^{(1)}$ and $k_S^{(2)}$ enter the measurement equation with an error term, and the residual diagnostics reported in the paper show that these error terms are well-behaved to such a degree that the statistical inference reported in the paper is valid.

**1.3.3 Plan to change paper in response**

We will include a discussion that relates the alternative measurements of the sink rate and AF to the budget imbalance, as explained above. When discussing residual diagnostics, we will point out the connection with the time series properties of the budget imbalance.

**Figure 6.** "Error" time series data

[Figure]

[Figure]

**1.4 Specific comment: 1**

**1.4.1 Referee comment**

*P1 L4 what do you mean by "balanced carbon budget" ?*

**1.4.2 Answer**

5    We mean that the sources of $CO_2$ should equal the sinks of $CO_2$, i.e., that the budget equation $E_t = G_t + S_t^L + S_t^O$ should hold (be "balanced") at all times and that any departures from this equation are due to measurement errors in the data. Departures from the equation are captured by the budget imbalance term, $B_t^{IM}$. Hence, what we mean is that this term is, on average, zero. (This has indeed been the case historically, see Le Quéré et al., 2018).

**1.4.3 Plan to change paper in response**

10    We will rephrase the sentence in the paper to better capture our intended meaning.

**1.5 Specific comment: 2**

**1.5.1 Referee comment**

*P1 L4 please clarify this sentence. It is unclear to me what object are you talking about*

**1.5.2 Answer**

We are specifically referring to the airborne fraction and the sink rate. Also notice that we give an example in parentheses at the end of the sentence: "(for example, the airborne fraction)".

**1.5.3 Plan to change paper in response**

We will rephrase the sentence in the paper to clearly identify the object we are referring to.

**1.6 Specific comment: 3**

**1.6.1 Referee comment**

*P1 L6 please explain a bit further because a decrease in the sink should end up ultimately by a change in the AF*

**1.6.2 Answer**

As explained in Section 8 of Gloor et al. (2010), it is not necessarily the case that a decrease in the sink rate implies an increase in the airborne fraction. We touch briefly on this in the Discussion section (P13 L 15). See also Raupach (2013).

The main point is that the airborne fraction is defined as $AF_t = G_t/E_t$, while the sink rate is defined as $k_{S,t} = S_t/C_t$. In other words, the normalizations of these time series are different, and they are not complements. Raupach et al. (2014) argue that the latter quantity is more appropriate as an object of study. However, due to the interest in the literature in both the sink rate as well as the airborne fraction, we have analyzed both quantities in the main paper. In the Discussion section we give some further arguments as to why the sink rate may be an easier object to analyze statistically than the airborne fraction (P13 L18).

**1.6.3 Plan to change paper in response**

We will add some clarifications on this in Section 2 of the main paper.

**1.7 Specific comment: 4**

**1.7.1 Referee comment**

*P1 L13 please add the reference period over which this % are estimated + the reference publication*

**1.7.2 Answer**

Thank you. We will do this in the revised version of the paper.

**1.8 Specific comment: 5**

**1.8.1 Referee comment**

*P1 L18 you could acknowledge more recent studies here*

**1.8.2 Answer**

5   Thanks for pointing this out, we will do so.

**1.9 Specific comment: 6**

**1.9.1 Referee comment**

*P2 L5 anthropic = anthropogenic*

**1.9.2 Answer**

10   Thank you. Corrected.

**1.10 Specific comment: 7**

**1.10.1 Referee comment**

*P2 L7 you can remove "which we argue is well designed for the problem at hand"*

**1.10.2 Answer**

15   Thank you. Removed.

**1.11 Specific comment: 8**

**1.11.1 Referee comment**

*P3 L12-16 I think paragraph should be move above and better explain why you are working on the "balanced" hypothesis. The Bim remains small compared to the other terms for example ?*

20 ### 1.11.2 Answer

Thanks for pointing this out. We will clarify this in the revised version.

**1.12 Specific comment: 9**

**1.12.1 Referee comment**

*P4 L2 could you further explain the meaning of "Using a simplifying linear specification ?*

**1.12.2 Answer**

5   In Section 3 of Gloor et al. (2010), the variable $k_{S,t}$ is interpreted as a "sink efficiency". To see why this is, note that we can write (cf. Equation (3) in the main paper)

$$S_t^O + S_t^L = k_{S,t} \cdot C_t.$$

In other words, $k_{S,t}$ is the amount of $CO_2$ transferred into the sinks, for every unit of $CO_2$ in the atmosphere above pre-industrial levels ($C_t$). In this way, $k_{S_t}$ gives an indication of the efficiency with which the carbon system transfers $CO_2$ to the

10   sinks. See also Raupach (2013) Section 3.1 for a discussion of this "efficiency" interpretation.

**1.12.3 Plan to change paper in response**

We will change the paper to better reflect the above discussion and make our statement clearer.

**1.13 Specific comment: 10**

**1.13.1 Referee comment**

15   *P6 L12-15 what about for a lower confidence threshold e.g., 90% do you get a better agreement ? why such a different in Beta estimates (one order of magnitude) ?*

**1.13.2 Answer**

In Table 1 of the main paper, we indeed get two different estimates of $\beta$, namely $0.00109$ and $0.00049$. However, we notice that the standard deviations of the estimates are given by $0.00179$ and $0.00203$, respectively. It indicates that although the estimates

20   are very different (by an order of magnitude, as pointed out by the referee), this difference is not statistically significant. The $p$-values are $0.5423$ and $0.8084$ respectively.

    The $p$-values also give an answer to the second question: the estimates are not significant at a $90\%$ level.

**1.13.3 Plan to change paper in response**

We will add the $p$-values to the main paper.

**1.14 Specific comment: 11**

**1.14.1 Referee comment**

*P7 L14 please give the estimate of TtA ? besides I think there is a error in Eq 13 with the random noise epsilon. I read it as independent of time.*

**1.14.2 Answer**

The estimate of $T_t^A$ is shown in Figure 1 in the main paper (page 8). The estimates of the accompanying parameters are given in Table 2 (page 7 in the main paper).

We have indeed missed the subscript in Equation (13). Thank you for pointing this out.

**1.14.3 Plan to change paper in response**

A subscript "t" will be added to the error term in Equation (13), P7 L14.

**1.15 Specific comment: 12**

**1.15.1 Referee comment**

*P10 L9-10 the last sentence requires further explanations.*

**1.15.2 Answer**

Agreed. The forecasts we provide are implied by the model and can be computed within our state space approach. The forecasts for the next 25 years are displayed in Fig. 3 of the main paper and the downward trend is the result of a negative estimate of $\beta$ as reported in Table 4. Under the current conditions, our forecast implies that it takes more than 25 years before the sink rate is below the value of $0.02$.

**1.15.3 Plan to change paper in response**

In the main paper, we will expand the explanation along the lines given here.

**1.16 Specific comment: 13**

**1.16.1 Referee comment**

*Figure 3 I don't know what these two panels show. They show the two metrics, correct ? Why giving the confidence interval for 1 sigma whereas most of the statistical test were conducted with a 95% confidence threshold ?*

**1.16.2 Answer**

Correct and agreed. We will explain more carefully. We will also give $95\%$ thresholds.

**1.17 Specific comment: 14**

**1.17.1 Referee comment**

5 *P12 L15 this looks like trivial. I guess that a simple correlation between the SF and LF should lead to the same conclusion. . .*

**1.17.2 Answer**

The wording of the sentence in the paper is somewhat unclear. What we meant to say is that the variation in the combined sink rate is mostly driven by the variation in the land sink rate. We will rephrase the paragraph to make the point clear.

**References**

Gloor, M., Sarmienti, J. L., and Gruber, N.: What can be learned about carbon cycle climate feedbacks from the CO2 airborne fraction?, Atmospheric Chemistry and Physics, 10, 7739 – 7751, 2010.

Le Quéré, C., Andrew, R. M., Friedlingstein, P., Sitch, S., Pongratz, J., Manning, A. C., Korsbakken, J. I., Peters, G. P., Canadell, J. G., Jackson, R. B., Boden, T. A., Tans, P. P., Andrews, O. D., Arora, V. K., Bakker, D. C. E., Barbero, L., Becker, M., Betts, R. A., Bopp, L., Chevallier, F., Chini, L. P., Ciais, P., Cosca, C. E., Cross, J., Currie, K., Gasser, T., Harris, I., Hauck, J., Haverd, V., Houghton, R. A., Hunt, C. W., Hurtt, G., Ilyina, T., Jain, A. K., Kato, E., Kautz, M., Keeling, R. F., Klein Goldewijk, K., Körtzinger, A., Landschützer, P., Lefèvre, N., Lenton, A., Lienert, S., Lima, I., Lombardozzi, D., Metzl, N., Millero, F., Monteiro, P. M. S., Munro, D. R., Nabel, J. E. M. S., Nakaoka, S.-I., Nojiri, Y., Padin, X. A., Peregon, A., Pfeil, B., Pierrot, D., Poulter, B., Rehder, G., Reimer, J., Rödenbeck, C., Schwinger, J., Séférian, R., Skjelvan, I., Stocker, B. D., Tian, H., Tilbrook, B., Tubiello, F. N., van der Laan-Luijkx, I. T., van der Werf, G. R., van Heuven, S., Viovy, N., Vuichard, N., Walker, A. P., Watson, A. J., Wiltshire, A. J., Zaehle, S., and Zhu, D.: Global Carbon Budget 2017, Earth System Science Data, 10, 405–448, https://doi.org/10.5194/essd-10-405-2018, https://www.earth-syst-sci-data.net/10/405/2018/, 2018.

Raupach, M. R.: The exponential eigenmodes of the carbon-climate sytem, and their implications for ratios of responses to forcings, Earth System Dynamics, 4, 31 – 49, 2013.

Raupach, M. R., Gloor, M., Sarmiento, J. L., Canadell, J. G., Frölicher, T. L., Gasser, T., Houghton, R. A., Le Quéré, C., and Trudinger, C. M.: The declining uptake rate of atmospheric $CO_2$ by land and ocean sinks, Biogeosciences, 11, 3453–3475, https://doi.org/10.5194/bg-11-3453-2014, https://www.biogeosciences.net/11/3453/2014/, 2014.

---

## Author Comment (AC2) · 30 Jan 2019

**Trend analysis of the airborne fraction and sink rate of anthropogenically released $CO_2$**

**Reply to referee report number 2**

Mikkel Bennedsen[1,3], Eric Hillebrand[1,3], and Siem Jan Koopman[2,3]

[1]Department of Economics and Business Economics, Aarhus University, Fuglesangs Allé, 4 8210 Aarhus V, Denmark
[2]Department of Econometrics, School of Business and Economics, Vrije Universiteit Amsterdam, De Boelelaan 1105, 1081 HV Amsterdam, The Netherlands.
[3]Center for Research in Econometric Analysis of Time Series (CREATES), Aarhus University, Fuglesangs Allé, 4 8210 Aarhus V, Denmark

**Correspondence:** Mikkel Bennedsen (mbennedsen@econ.au.dk)

**1 Introduction**

We thank the referee for the insightful comments and the supportive review. We will revise the paper in response to your comments, and we think that the paper will improve substantially as a result.

**1.1 Comment 1**

**1.1.1 Referee comment**

*Firstly in the introduction the authors state: " a key question is whether the airborne fraction is increasing " but they do not say why. It would be good if they would add why this is so.*

**1.1.2 Answer**

We will add some text in the paper explaining the importance and add some references to the literature such as Gloor et al. (2010), Raupach et al. (2014), Bacastow and Keeling (1979), Schimel et al. (2001).

**1.2 Comment 2**

**1.2.1 Referee comment**

*Sentence just above - 24% and 31% - I would add a reference here - and possibly uncertainties - just for completeness.*

**1.2.2 Answer**

We will add a reference to Le Quéré et al (2018). (We calculated these numbers from the GCB data.) Similar numbers have been reported elsewhere, e.g., Ballantyne et al. (2012).

**1.3 Comment 3**

**1.3.1 Referee comment**

*When applying the Kalman filter the authors will need to initialize it. I may have missed it but if not it would be good if the authors would add this in the main text.*

**1.3.2 Answer**

We use a diffuse initialisation of the Kalman Filter as outlined in Chapter 5 of Durbin and Koopman (2012). We will make this clear in the main text in Section 2.

**1.4 Comment 3**

**1.4.1 Referee comment**

*Finally my comment - while all the results are sound - what the paper does not explain is the true reason for the decrease in sink rates - and thus it is not clear whether a decreasing sink rate is alarming or not. It would be nice if the authors could comment on that - but it is not a necessary condition.*

*Some earlier papers actually give a clue what the real reason may be.*

**1.4.2 Answer**

Thank you for pointing this out. Raupach (2013) argue that a necessary condition for a constant sink rate is that emissions ($E_t$) grow exponentially. Hence, a decreasing sink rate could be the result of less-than-exponential growth in emissions.

Another explanation can be fertilisation/saturation of the sinks. To illustrate this, we focus on the land sink rate, since we find some evidence in the paper for a decreasing land sink rate. Recall that (Equation (5) in the main paper)

$$S_t^L = k_{L,t} \cdot C_t,$$

where $k_{L,t}$ is the land sink rate, $S_t^L$ the land sink $CO_2$ flux, and $C_t$ the amount of $CO_2$ in the atmosphere above pre-industrial levels. If the flux of $CO_2$ to the land sink was linear in $C_t$, then $k_{L,t}$ would be constant. Conversely, a decreasing $k_{L,t}$ implies that the efficiency with which the land sink absorbs $CO_2$ is decreasing. That is, the flux of $CO_2$ to the land sink is non-linear in $C_t$ and this non-linearity is such that the efficiency is decreasing. This is in line with simulation results from climate cycle models (Friedlingstein et al., 2006).

We can illustrate how such non-linearities can arise. The precise relationship between $S_t^L$ and $C_t$ still alludes us but Bacastow and Keeling (1973) (p. 94) suggest that (in our notation):

$$S_t^L \approx \beta \log(1 + C_t/\mathcal{C}^0),$$

wherre $\mathcal{C}^0 = 591.30$ GtC is the amount of $CO_2$ in the atmosphere in pre-industrial times. Using this, we can deduce

5  $S_t^L \approx \beta \log(1 + C_t/\mathcal{C}^0)$

$$\approx \beta \frac{C_t}{\mathcal{C}^0} - \frac{1}{2}\beta \left(\frac{C_t}{\mathcal{C}^0}\right)^2.$$

Now, if $\mathcal{C}^0$ is large as compared to $C_t$, this shows how a linear specification between $S_t^L$ and $C_t$ might be reasonable. However, once $C_t$ becomes large as compared to $\mathcal{C}^0$, this shows how the the estimated sink rate can be found to be decreasing. To see this, use the above to write

10  $S_t^L \approx k_{L,t} C_t,$

where

$$k_{L,t} = \frac{\beta}{\mathcal{C}^0} - \frac{1}{2}\frac{\beta}{\mathcal{C}^0}\frac{C_t}{\mathcal{C}^0}$$

is decreasing in $C_t$. For example, we have $C_{1959} \approx 80$ GtC and $C_{2016} \approx 267$ GtC, resulting in $C_{1959}/\mathcal{C}^0 \approx 14\%$ and $C_{2016}/\mathcal{C}^0 \approx 45\%$.

15  ### 1.4.3  Plan to change paper in response

We will add the references and the discussion to the main paper.

**References**

Bacastow, R. and Keeling, C. D.: Atmospheric Carbon Dioxide and radiocarbon in the natural cycle: II. Changes from A. D. 1700 to 2070 as deduced from a geochemical model, in: Carbon and the biosphere conference proceedings; Upton, New York, USA, 1973.

Bacastow, R. B. and Keeling, C. D.: Models to predict future atmospheric CO2 concentrations, in: Workshop on the global effects of carbon dioxide from fossil fuels, pp. 72–90, US Department of Energy, 1979.

Ballantyne, A. P., Alden, C. B., Miller, J. B., Tans, P. P., and White, J. W. C.: Increase in observed net carbon dioxide uptake by land and oceans during the past 50 years, Nature, 488, 70 EP –, https://doi.org/10.1038/nature11299, 2012.

Durbin, J. and Koopman, S. J.: Time series analysis by state space methods, 38, Oxford University Press, 2012.

Friedlingstein, P., Cox, P., Betts, R., Bopp, L., von Bloh, W., Brovkin, V., Cadule, P., Doney, S., Eby, M., Fung, I., Bala, G., John, J., Jones, C., Joos, F., Kato, T., Kawamiya, M., Knorr, W., Lindsay, K., Matthews, H. D., Raddatz, T., Rayner, P., Reick, C., Roeckner, E., Schnitzler, K.-G., Schnur, R., Strassmann, K., Weaver, A. J., Yoshikawa, C., and Zeng, N.: Climate–Carbon Cycle Feedback Analysis: Results from the C4MIP Model Intercomparison, Journal of Climate, 19, 3337–3353, https://doi.org/10.1175/JCLI3800.1, https://doi.org/10.1175/JCLI3800.1, 2006.

Gloor, M., Sarmienti, J. L., and Gruber, N.: What can be learned about carbon cycle climate feedbacks from the CO2 airborne fraction?, Atmospheric Chemistry and Physics, 10, 7739 – 7751, 2010.

Raupach, M. R.: The exponential eigenmodes of the carbon-climate sytem, and their implications for ratios of responses to forcings, Earth System Dynamics, 4, 31 – 49, 2013.

Raupach, M. R., Gloor, M., Sarmiento, J. L., Canadell, J. G., Frölicher, T. L., Gasser, T., Houghton, R. A., Le Quéré, C., and Trudinger, C. M.: The declining uptake rate of atmospheric $CO_2$ by land and ocean sinks, Biogeosciences, 11, 3453–3475, https://doi.org/10.5194/bg-11-3453-2014, https://www.biogeosciences.net/11/3453/2014/, 2014.

Schimel, D. S., House, J. I., Hibbard, K. A., Bousquet, P., Ciais, P., Peylin, P., Braswell, B. H., Apps, M. J., Baker, D., Bondeau, A., Canadell, J., Churkina, G., Cramer, W., Denning, A. S., Field, C. B., Friedlingstein, P., Goodale, C., Heimann, M., Houghton, R. A., Melillo, J. M., Moore III, B., Murdiyarso, D., Noble, I., Pacala, S. W., Prentice, I. C., Raupach, M. R., Rayner, P. J., Scholes, R. J., Steffen, W. L., and Wirth, C.: Recent patterns and mechanisms of carbon exchange by terrestrial ecosystems, Nature, 414, 169 EP –, https://doi.org/10.1038/35102500, 2001.

---

## Author Response (AR1)

**Trend analysis of the airborne fraction and sink rate of anthropogenically released $CO_2$**

**Author response, Review 1**

Mikkel Bennedsen[1,3], Eric Hillebrand[1,3], and Siem Jan Koopman[2,3]

[1]Department of Economics and Business Economics, Aarhus University, Fuglesangs Allé, 4 8210 Aarhus V, Denmark
[2]Department of Econometrics, School of Business and Economics, Vrije Universiteit Amsterdam, De Boelelaan 1105, 1081 HV Amsterdam, The Netherlands.
[3]Center for Research in Econometric Analysis of Time Series (CREATES), Aarhus University, Fuglesangs Allé, 4 8210 Aarhus V, Denmark

**Correspondence:** Mikkel Bennedsen (mbennedsen@econ.au.dk)

Sections 1 and 2 contain the original comments from referee 1 and referee 2, respectively. Sections 3 and 4 present our responses. Section 5 is a change-log, documenting the changes made to the main paper. At the end of this document, the revised version of the paper is reproduced with the changes indicated.

**1 Referee report number 1**

5 Below are the comments from the first referee.

**1.1 Comment 1**

*First, this study comes after almost a decade-long research (Knorr (2009), Gloor et al. (2010), and Ballantyne et al. (2015)) on the detection of the changes in AF or sink efficiency and does not provide new findings (e.g., results are in the line of Raupach 2014). Yet this work merits to be acknowledged because it is the first to my knowledge to investigate this long debate*

10 *on the stationarity of the AF or SF variations. Here the authors confirm that there is no non-stationnarity in AF and SF using GCP2018 data (from 1959 to 2017). Therefore, I am wondering if it is not the real outcomes of the study ? I mean once the stationarity of the variance is proved, the state space system loses some interest. The potential caveats as suggested by Gloor et al 2010 are removed and thus a simple linear model can be used to estimate trends in AF and SF. Standard statistics can be then used to detect if the signal (the trends) is larger than the noise (the variability).*

15 ### 1.2 Comment 2

*The second major comment concerns the attribution of the decreasing sink to the land carbon sink. Regarding the shape of the land C sink, we may be interested to test since how many years the land sink has started to decrease. To further this comment,*

*I think that several test of the length of the data and the influence of the sampling are missing in the manuscript. We need to see how far this approach is robust when using, for example, 5-year average data (removing ENSO and volcanoes influence).*

**1.3 Comment 3**

*My last major comment relates to the use of the "balanced" C budget whereas Le Quere et al. 2018 provides the Bim terms that could be used as a third entry in you model. I mean does the variance of Bim is steady in time or does it vary ? How far this terms correlates with AF and SF ? Do you fin a trends in Bim that could explain why the sink rate declines whereas the AF does ? I think all these discussions might consolidate the study.*

**1.4 Specific comment: 1**

*P1 L4 what do you mean by "balanced carbon budget" ?*

**1.5 Specific comment: 2**

*P1 L4 please clarify this sentence. It is unclear to me what object are you talking about*

**1.6 Specific comment: 3**

*P1 L6 please explain a bit further because a decrease in the sink should end up ultimately by a change in the AF*

**1.7 Specific comment: 4**

*P1 L13 please add the reference period over which this % are estimated + the reference publication*

**1.8 Specific comment: 5**

*P1 L18 you could acknowledge more recent studies here*

**1.9 Specific comment: 6**

*P2 L5 anthropic = anthropogenic*

**1.10 Specific comment: 7**

*P2 L7 you can remove "which we argue is well designed for the problem at hand"*

**1.11 Specific comment: 8**

*P3 L12-16 I think paragraph should be move above and better explain why you are working on the "balanced" hypothesis. The Bim remains small compared to the other terms for example ?*

**1.12   Specific comment: 9**

*P4 L2 could you further explain the meaning of "Using a simplifying linear specification ?*

**1.13   Specific comment: 10**

*P6 L12-15 what about for a lower confidence threshold e.g., 90% do you get a better agreement ? why such a different in Beta estimates (one order of magnitude) ?*

**1.14   Specific comment: 11**

**1.14.1   Referee comment**

*P7 L14 please give the estimate of TtA ? besides I think there is a error in Eq 13 with the random noise epsilon. I read it as independent of time.*

**1.15   Specific comment: 12**

*P10 L9-10 the last sentence requires further explanations.*

**1.16   Specific comment: 13**

*Figure 3 I don't know what these two panels show. They show the two metrics, correct ? Why giving the confidence interval for 1 sigma whereas most of the statistical test were conducted with a 95% confidence threshold ?*

**1.17   Specific comment: 14**

*P12 L15 this looks like trivial. I guess that a simple correlation between the SF and LF should lead to the same conclusion. . .*

**2 Referee report number 2**

Below are the comments from the second referee.

**2.1 Comment 1**

*Firstly in the introduction the authors state: " a key question is whether the airborne fraction is increasing " but they do not say why. It would be good if they would add why this is so.*

**2.2 Comment 2**

*Sentence just above - 24% and 31% - I would add a reference here - and possibly uncertainties - just for completeness.*

**2.3 Comment 3**

*When applying the Kalman filter the authors will need to initialize it. I may have missed it but if not it would be good if the authors would add this in the main text.*

**2.4 Comment 4**

*Finally my comment - while all the results are sound - what the paper does not explain is the true reason for the decrease in sink rates - and thus it is not clear whether a decreasing sink rate is alarming or not. It would be nice if the authors could comment on that - but it is not a necessary condition.*

*Some earlier papers actually give a clue what the real reason may be.*

**3 Response to referee report number 1**

**3.1 Response to Comment 1 (Section 1.1)**

Thank you for raising these important points. In our approach, stationarity or non-stationarity is a *finding* rather than an *assumption*. We agree with the referee that stationarity of the AF and negative linear trending in the sink rate are the main results, *ex post*. However, *a priori*, we have formulated a statistical dynamic model that allows for both, non-stationary and stationary processes for $y_t$ and for linear as well as stochastic trending behavior. In the paper, below equation (7), we show that the solution to the difference equation (7) leads to a deterministic time trend when the iid Gaussian random variable $\eta_t$ is zero (effectively when its variance is zero, that is $\sigma_\eta^2 = 0$). In this case, the time series is *trend-stationary* and the dynamic process for $y_t$ does not exhibit a unit-root (which is the case for a *difference-stationary* time series). Since $\sigma_\eta^2$ is estimated using the observations for $y_t$, we can only conclude *ex post* that the time series is trend-stationary. Without the estimation of our state-space model, we could not have arrived at this conclusion.

Our main findings can be summarized as follows. (a) We find no statistical evidence of an increasing airborne fraction, while we do find statistical evidence for a decreasing sink rate. (b) While the findings (a) have also been reported elsewhere, most notably in Raupach et al. (2014), our statistical model does not make any a priori assumptions regarding the stationarity or non-stationary of the series and regarding deterministic or stochastic behaviour of the trends. These findings are thus *results*, as opposed to assumptions, of our approach. Furthermore, we find that we need to estimate our model on *all* the data from the global carbon budget jointly to reach the findings (a).

In the joint estimation, we take all data into account, that is the time series for AF and SR as defined on page 3 of the paper, but also the additional data obtained by assuming that the carbon budget is balanced, which we explain on page 4 of the paper. (Note that we follow the sink rate definition of Raupach (2013) and Raupach et al. (2014), with concentrations in the denominator, not emissions. The sink rate (SR) in our paper is thus not the complement of the airborne fraction.) More specifically, when analyzing the sink fraction, for example, we can opt for either one or both of the two time series:

$$k_S^{(1)} = \frac{S_t^O + S_t^L}{C_t},$$
$$k_S^{(2)} = \frac{E_t - G_t}{C_t},$$

where we exploit the carbon budget equation to obtain $k_S^{(2)}$:

$$E_t = G_t + S_t^O + S_t^L + B_t^{IM}.$$

The budget imbalance $B_t^{IM}$ is a zero-mean noise sequence that represents the measurement errors in the other variables of the carbon budget (Le Quéré et al., 2018). In Table 3 of the main paper we report the results when we estimate our state-space model on $k_S^{(1)}$ and $k_S^{(2)}$ separately. In Table 4 and Figure 2 of the main paper we report the results when we estimate the state-space model on $k_S^{(1)}$ and $k_S^{(2)}$ jointly. It is a feature of the state-space model that it allows for alternative measurements for the same object of interest.

In the Supplemental Material file, we have included Figure 1 that presents the extracted the latent trends, $T_t$ in equation (7), from the separate analysis, and Figure 2 that presents the extracted comment trend $T_t$ from the joint analysis. (This is a replication of Figure 2 in the main paper.) The extraction method is based on the filtering and smoothing approach as discussed in the main paper. The various extracted trends illustrate our points as argued above: in the separate analysis, we obtain (slightly) time-varying stochastic trends, whereas in the joint analysis, we obtain a deterministic linear trend with a significantly negative slope (compare Panel B of Table 4 in the main paper). We have found similar results for the AF series, which we include in the Supplemental Material file.

**Figure 1.** Univariate estimation of time-varying trend for sink rate, $k_S$.

[Figure]

**3.1.1 Changes made to the paper in response**

We have re-written the explanations of the AF and SR definitions in Sect. 2 to show the contribution of the state-space model in the context more crisply.

We have included Figure 1 and the corresponding results for the AF in a Supplementary Material file.

**3.2 Response to Comment 2 (Section 1.2)**

The referee raises two interesting questions; we treat them one-by-one.

(i) With respect to the land sink rate, we are treating it as a fraction: the ratio of flux in land sink over $CO_2$ concentration in the atmosphere. The land sink flux itself, $S_t^L$, is increasing over time but the land sink rate, $k_{L,t} = S_t^L/C_t$, shows evidence of a decreasing trend. Our paper shows (Figure 2 in this reply) that if we sum up ocean and land sink and use both time series for this object, $k_S^{(1)}$ and $k_S^{(2)}$, jointly, we obtain a significantly negatively sloping deterministic trend. We can of course also

**Figure 2.** Multivariate estimation of time-varying trend for sink rate, $k_S$.

[Figure]

consider $k_S^{(1)}$ and $k_S^{(2)}$ separately. The result is shown in Figure 1 of this reply. From the left panel of this figure, one might argue that the negative trend started in the mid 1970s. The right panel, which shows $k_S^{(2)}$, however, does not display such a kink. Finally, we can consider the land sink rate, $k_L$, individually. The result is shown in Figure 3 of this reply. We obtain a deterministic trend with an insignificant negative slope, cf. Table 5 of the main paper.

5    (ii) We have estimated the state-space model on 5-year average data in order to reduce the impact of effects such as ENSO, volcanic eruptions, and the like. The state-space model estimated on annual data is also capable of accounting for these effects, since it treats them as additive noise in the measurement equation. Repeating the analysis based on 5-year average data, however, provides a way to verify our estimation results and conclusions. (We also considered 2-, 3-, and 4-year averages, with similar results.)

10    We calculate 5-year non-overlapping averages in order to avoid introducing serial correlation into the time series. Running (i.e., overlapping) averages would necessitate specifying a model to capture this serial correlation, and we think this is a relatively bigger disadvantage than the reduction in the sample size that we incur from non-overlapping averages. Since we have 58 years of data, we calculate an average of the first three years followed by 5-year averages, resulting in 12 observations. The findings from estimating the state-space model on these time series of averages confirm those reported in the main paper:

15  In the joint estimation, we find no statistical evidence of a trend in the airborne fraction (with a $p$-value of $0.32138$), and we do find statistical evidence of a decreasing trend in the sink rate (with a $p$-value of $0.00064$). Of course, the residual diagnostics for these short time series are not as convincing as those presented in the main paper. The extracted trends from these joint analyses are presented in Figures 4 (airborne fraction) and 5 (sink rate) in this reply. Incidentally, some analyses in earlier studies were based on running averages; we discuss these briefly in the Discussion section of the main paper (P13 L10).

We emphasize two points in this context: (1) The state-space model is advantageous in this exercise, since it allows to incorporate the alternative time series for both, AF and SR, which is particularly useful when the sample period is short. (2) The main finding from annual data prevails: In the separate analyses, the trends are estimated as stochastic. Only in the joint analysis do we obtain a deterministic trend and statistical significance for the sink rate.

**Figure 3.** Univariate estimation of time-varying trend for land sink rate, $k_L$.

[Figure]

**Figure 4.** Multivariate estimation of time-varying trend for AF. Data: 5-year averages (no overlap)

[Figure]

**Figure 5.** Multivariate estimation of time-varying trend for sink rate, $k_S$. Data: 5-year averages (no overlap)

**3.2.1 Changes made to the paper in response**

- (i) We include the separate analysis of the land sink rate ($k_L$) in the Supplementary Material. For completeness, we also submit the ocean sink rate ($k_O$) to the same analysis in the Supplementary Material.

- (ii) We include the estimation results based on 5-year averages in the Supplementary Material and briefly discuss these findings in the main paper as well, with a reference to the Supplementary Material for further details (cf. the Discussion Sect.). We emphasize that the findings of the paper are robust to averaging of the data.

**3.3 Response to Comment 3 (Section 1.3)**

Thank you for raising this point. By assuming that the carbon budget is balanced, we already include the $B_t^{IM}$ data in the analysis. Specifically, the data on the budget imbalance enters as follows. For the case of the sink rate, the two time series employed are:

$$k_S^{(1)} = \frac{S_t^O + S_t^L}{C_t},$$
$$k_S^{(2)} = \frac{E_t - G_t}{C_t}.$$

Given the carbon budget equation, the latter expression can be written as

$$k_S^{(2)} = \frac{E_t - G_t}{C_t} = \frac{S_t^O + S_t^L + B_t^{IM}}{C_t} = k_S^{(1)} + \xi_t,$$

where

$$\xi_t = \frac{B_t^{IM}}{C_t},$$

can be regarded as an error term. This is the motivation for using the two time series $k_S^{(1)}$ and $k_S^{(2)}$ as data for the same underlying quantity, that is, the sink rate.

Figure 6 plots the time series of $\xi_t$ (left plot) and $B_t^{IM}$ (right plot). Both have a mean that is not significantly different from zero and follow stationary dynamics, albeit with some serial correlation.

In the joint state-space model, both $k_S^{(1)}$ and $k_S^{(2)}$ enter the measurement equation with an error term, and the residual diagnostics reported in the paper show that these error terms are well-behaved to such a degree that the statistical inference reported in the paper is valid.

**Figure 6.** "Error" time series data

[Figure]

[Figure]

**3.3.1 Changes made to the paper in response**

We include a discussion that relates the alternative measurements of the sink rate and AF to the budget imbalance, as explained above. When discussing residual diagnostics, we point out the connection with the time series properties of the budget imbalance.

**3.4 Response to Specific Comment 1 (Section 1.4)**

We mean that the sources of $CO_2$ should equal the sinks of $CO_2$, i.e., that the budget equation $E_t = G_t + S_t^L + S_t^O$ should hold (be "balanced") at all times and that any departures from this equation are due to measurement errors in the data. Departures

from the equation are captured by the budget imbalance term, $B_t^{IM}$. Hence, what we mean is that this term is, on average, zero. (This has indeed been the case historically, see Le Quéré et al., 2018).

**3.4.1 Changes made to the paper in response**

We have rephrased the sentence in the paper to better capture our intended meaning.

**3.5 Response to Specific Comment 2 (Section 1.5)**

We are specifically referring to the airborne fraction and the sink rate. Also notice that we give an example in parentheses at the end of the sentence: "(for example, the airborne fraction)".

**3.5.1 Changes made to the paper in response**

We have rephrased the sentence in the paper to clearly identify the object we are referring to.

**3.6 Response to Specific Comment 3 (Section 1.6)**

As explained in Section 8 of Gloor et al. (2010), it is not necessarily the case that a decrease in the sink rate implies an increase in the airborne fraction. We touch briefly on this in the Discussion section of the initial submission (P13 L 15). See also Raupach (2013).

The main point is that the airborne fraction is defined as $AF_t = G_t/E_t$, while the sink rate is defined as $k_{S,t} = S_t/C_t$. In other words, the normalizations of these time series are different, and they are not complements. Raupach et al. (2014) argue that the latter quantity is more appropriate as an object of study. However, due to the interest in the literature in both the sink rate as well as the airborne fraction, we have analyzed both quantities in the main paper. In the Discussion section we give some further arguments as to why the sink rate may be an easier object to analyze statistically than the airborne fraction (P13 L18).

**3.6.1 Changes made to the paper in response**

We have rewritten much of Sect. 2 of the paper to make things more clear.

**3.7 Response to Specific Comment 4 (Section 1.7)**

Thank you. We have done this in the revised version of the paper.

**3.8 Response to Specific Comment 5 (Section 1.8)**

Thanks for pointing this out, we have done so in the revised version of the paper.

**3.9 Response to Specific Comment 6 (Section 1.9)**

Thank you. Corrected.

**3.10 Response to Specific Comment 7 (Section 1.10)**

Thank you. Removed.

**3.11 Response to Specific Comment 8 (Section 1.11)**

Thanks for pointing this out. We have clarified this in the revised version of the paper.

**3.12 Response to Specific Comment 9 (Section 1.12)**

In Section 3 of Gloor et al. (2010), the variable $k_{S,t}$ is interpreted as a "sink efficiency". To see why this is, note that we can write (cf. Equation (3) in the main paper)

$$S_t^O + S_t^L = k_{S,t} \cdot C_t.$$

In other words, $k_{S,t}$ is the amount of $CO_2$ transferred into the sinks, for every unit of $CO_2$ in the atmosphere above pre-industrial levels ($C_t$). In this way, $k_{S_t}$ gives an indication of the efficiency with which the carbon system transfers $CO_2$ to the sinks. See also Raupach (2013) Section 3.1 for a discussion of this "efficiency" interpretation.

**3.12.1 Changes made to the paper in response**

We have changed the paper to better reflect the above discussion and make our statement clearer.

**3.13 Response to Specific Comment 10 (Section 1.13)**

In Table 1 of the main paper, we indeed get two different estimates of $\beta$, namely $0.00109$ and $0.00049$. However, we notice that the standard deviations of the estimates are given by $0.00179$ and $0.00203$, respectively. It indicates that although the estimates are very different (by an order of magnitude, as pointed out by the referee), this difference is not statistically significant. The $p$-values are $0.5423$ and $0.8084$ respectively.

The $p$-values also give an answer to the second question: the estimates are not significant at a $90\%$ level.

**3.13.1 Changes made to the paper in response**

We have added the $p$-values to the main paper.

**3.14 Response to Specific Comment 11 (Section 1.14)**

The estimate of $T_t^A$ is shown in Figure 1 in the main paper (page 8, initial version). The estimates of the accompanying parameters are given in Table 2 (page 7 in the initial version of the main paper).

We have indeed missed the subscript in Equation (13). Thank you for pointing this out.

**3.14.1 Changes made to the paper in response**

Subscript "t"'s have been added in the equations where they were missing.

**3.15 Response to Specific Comment 12 (Section 1.15)**

5   Agreed. The forecasts we provide are implied by the model and can be computed within our state space approach. The forecasts for the next 25 years are displayed in Fig. 3 of the main paper and the downward trend is the result of a negative estimate of $\beta$ as reported in Table 4. Under the current conditions, our forecast implies that it takes more than 25 years before the sink rate is below the value of $0.02$.

**3.15.1 Changes made to the paper in response**

10   We have added some additional comments on the forecasting exercise.

**3.16 Response to Specific Comment 13 (Section 1.16)**

Correct and agreed. We now explain this more carefully. We have also given $95\%$ thresholds instead of $68\%$ thresholds.

**3.17 Response to Specific Comment 14 (Section 1.17)**

The wording of the sentence in the paper is somewhat unclear. What we meant to say is that the variation in the combined sink
15   rate is mostly driven by the variation in the land sink rate. We have rephrased the paragraph to make the point clear.

**4 Response to referee report number 2**

**4.1 Response to Comment 1 (Section 2.1)**

We have added some text in the paper explaining the importance and added some references to the literature (e.g., Gloor et al. (2010), Raupach et al. (2014), Bacastow and Keeling (1979)).

**4.2 Response to Comment 2 (Section 2.2)**

We calculated these numbers from the GCB data. We have included this information, along with a reference where similar numbers can be found.

**4.3 Response to Comment 3 (Section 2.3)**

We use a diffuse initialisation of the Kalman Filter as outlined in Chapter 5 of Durbin and Koopman (2012). We added a comment on this in the main paper.

**4.4 Response to Comment 4 (Section 2.4)**

Thank you for pointing this out. Raupach (2013) argue that a necessary condition for a constant sink rate is that emissions ($E_t$) grow exponentially. Hence, a decreasing sink rate could be the result of less-than-exponential growth in emissions.

Another explanation can be fertilisation/saturation of the sinks. To illustrate this, we focus on the land sink rate, since we find some evidence in the paper for a decreasing land sink rate. Recall that (Equation (5) in the main paper)

$$S_t^L = k_{L,t} \cdot C_t,$$

where $k_{L,t}$ is the land sink rate, $S_t^L$ the land sink $CO_2$ flux, and $C_t$ the amount of $CO_2$ in the atmosphere above pre-industrial levels. If the flux of $CO_2$ to the land sink was linear in $C_t$, then $k_{L,t}$ would be constant. Conversely, a decreasing $k_{L,t}$ implies that the efficiency with which the land sink absorbs $CO_2$ is decreasing. That is, the flux of $CO_2$ to the land sink is non-linear in $C_t$ and this non-linearity is such that the efficiency is decreasing. This is in line with simulation results from climate cycle models (Friedlingstein et al., 2006).

We can illustrate how such non-linearities can arise. The precise relationship between $S_t^L$ and $C_t$ still alludes us but Bacastow and Keeling (1973) (p. 94) suggest that (in our notation):

$$S_t^L \approx \beta \log(1 + C_t/\mathcal{C}^0),$$

wherre $\mathcal{C}^0 = 591.30$ GtC is the amount of $CO_2$ in the atmosphere in pre-industrial times. Using this, we can deduce

$$
\begin{aligned}
S_t^L &\approx \beta \log(1 + C_t/\mathcal{C}^0) \\
&\approx \beta \frac{C_t}{\mathcal{C}^0} - \frac{1}{2}\beta \left( \frac{C_t}{\mathcal{C}^0} \right)^2 .
\end{aligned}
$$

Now, if $\mathcal{C}^0$ is large as compared to $C_t$, this shows how a linear specification between $S_t^L$ and $C_t$ might be reasonable. However, once $C_t$ becomes large as compared to $\mathcal{C}^0$, this shows how the the estimated sink rate can be found to be decreasing. To see this, use the above to write

$$S_t^L \approx k_{L,t} C_t,$$

where

$$k_{L,t} = \frac{\beta}{\mathcal{C}^0} - \frac{1}{2} \frac{\beta}{\mathcal{C}^0} \frac{C_t}{\mathcal{C}^0}$$

is decreasing in $C_t$. For example, we have $C_{1959} \approx 80$ GtC and $C_{2016} \approx 267$ GtC, resulting in $C_{1959}/\mathcal{C}^0 \approx 14\%$ and $C_{2016}/\mathcal{C}^0 \approx 45\%$.

**4.4.1 Changes made to the paper in response**

We have changed the paper in several places to better explore these important questions (cf. the change-log below). In particular, we have added the mathematical derivations above to Appendix A.

**5   Change-log**

Below we list the changes made in the revised version of the paper. This version is reproduced at the end of this document with changes indicated by colour; red for deletions and blue for additions. All page and line numbers refer to this new version reproduced below.

**5.1   Changes**

The following reported changes are organized in the order of the questions posed by the referees, reproduced above. Each change is given a number, which we refer to as a "Point". For instance, if we want to refer back to the first change described, we would refer to "Point 1".

Points 1–3 report changes of a general nature; Point 4 is in response to a comment from the Associate Editor; Points 5–21 are in response to the comments from referee 1; and Points 22–25 are in response to the comments from referee 2.

1. We have included a Supplementary Material containing additional statistical analyses. We reference this in the Introduction (P2, L34) and the Discussion Sect. 7 (P17, L16).

2. We have added $p$-values for the different hypothesis tests conducted, see P8, L1 for an example.

3. We have corrected the wording and typos in several places, as highlighted by the tracked changes.

4. We provide a more in-depth discussion of why the topic of this article is an important area of study (P1, first paragraph of introduction). We have also sought to contextualise the paper better: We emphasize more that our approach addresses the methodological criticism of earlier studies (P2, L22-28); we work out better that our findings of deterministic trends are *results*, as opposed to being a priori assumptions as in earlier studies (P15, L13-15); we have developed further our investigation of the apparent decreasing land sink and proposed possible explanations for this (P2, L27; P16, L29; Appendix A, P19). This latter point connects to a large literature on the behavior of the terrestrial land sink. (P17, L1; P18, L10-13; P19, L3-4, P19, L11) This is in response to comments from the Associate Editor.

5. P3-P5: We have cleared up the definitions of the airborne fraction and the sink rate in Sect. 2. This is in response to the comment in Section 1.1.

6. P17, L3: In the Discussion section, we present the results from the analysis using 5-year average data and reference the Supplementary Material, where the details of this analysis can be found. This is in response to the comment in Section 1.2.

7. In response to the comment in Section 1.3, we have changed the following:

    (a) P7, L10, Eq. (8): We have changed the notation slightly, when introducing the two different data series for the AF.

    (b) P10, L14, Eq. (13): We have changed the notation slightly, when introducing the two different data series for the SR.

(c) P8, L13-15: We briefly discuss the statistical properties of the error term $\xi_t = B_t^{IM}/E_t^{ANT}$ which are implied by the diagnostics in Table 1.

(d) P11, L5-7: We briefly discuss the statistical properties of the error term $\xi_t = B_t^{IM}/C_t$ which are implied by the diagnostics in Table 1.

8. P1, L4: We changed "balanced carbon budget" to capture our intended meaning. This is in response to the comment in Section 1.4.

9. P1, L5: We re-phrased this part to make things more clear. This is in response to the comment in Section 1.5.

10. P5, L4: We comment on the differences between the airborne fraction and the sink rate. This is in response to the comment in Section 1.6.

11. P1. L15: We added the period over which the numbers were calculated as well as references. For some formatting reason, this addition can not be read in the version which tracks the changes (it reads fine in the revised version of the paper). The sentence reads: "These percentages are calculated over the period 1959 to 2016 using the data described below, see e.g. Raupach et al. (2014) for similar estimates.". This is in response to the comment in Section 1.7.

12. P2, L6: We added the reference "Rayner et al. (2015)". This is in response to the comment in Section 1.8.

13. P2, L13: anthropic → anthropogenic (although this discussion has been changed). In response to the comment in Section 1.9.

14. P2, L16: Removed "which we argue is well designed for the problem at hand". In response to the comment in Section 1.10.

15. P3, L15-17: We discuss the budget imbalance term $B_t^{IM}$ a bit more in-depth, motivating it's later role as part of an error term. The justification for this is strengthened by the diagnostics coming from the state-space analysis, cf. the changes discussed above in points 7c and 7d. This is in response to the comment in Section 1.11.

16. P5, L8: We have deleted the part starting with "Using a simplifying linear specification...". Indeed, the whole of Sect. has been re-worked to make our intended meaning more clear, cf. also point 10 above. This is in response to the comment in Section 1.12.

17. As mentioned above in point 2, we have added p-values to the estimates of the slope parameters. This is partly in response to the comment in Section 1.13.

18. A subscript $t$ has been added in Eq. (13) (which is now Eq. (12) on P9) and others where it was missing. This is response to the comment in Section 1.14.

19. p. 12, L10: We elaborate on the forecasting exercise. This is in response to the comment in Section 1.15.

20. P13, Figure 3. We explained this figure more in-depth in the text, cf. the comment in Point 19. We also substituted the 68% confidence bands for 95% confidence bands in Figure 3 in the main paper. This is in response to the comment in Section 1.16.

21. P14, L20: The wording of last sentence in the last paragraph of Sect. 6 has been changed slightly. This is in response to the comment in Section 1.17.

22. P1 (first paragraph of introduction): We elaborate on why the topic of this paper is important, cf. also Point 4 above. This is in response to the comment in Section 2.1. Cf. also Point 4.

23. P1, L15: We added a comment on how we arrived at those numbers (45%, 24%, 31%) plus a reference. For some formatting reason, this addition can not be read in the version which tracks the changes (it reads fine in the revised version of the paper). The sentence reads: "These percentages are calculated over the period 1959 to 2016 using the data described below, see e.g. Raupach et al. (2014) for similar estimates.". This is in response to the comment in Section 2.2. Cf. also Point 11.

24. P6, L7-8: We have added a comment on the initialization of the Kalman filter. This is in response to the comment in Section 2.3.

25. P16, L29: We have added a paragraph in the Discussion section, where we offer possible explanations for the finding of the decreasing sink rate, see also Appendix A (P19). Similarly, we now briefly comment on this in the Conclusion (P18, L10-13). This is in response to the comment in Section 2.4.

[revised manuscript text omitted]

---

## Author Response (AR2)

**Trend analysis of the airborne fraction and sink rate of anthropogenically released $CO_2$**

**Author response, Review 2**

Mikkel Bennedsen[1,3], Eric Hillebrand[1,3], and Siem Jan Koopman[2,3]

[1]Department of Economics and Business Economics, Aarhus University, Fuglesangs Allé, 4 8210 Aarhus V, Denmark
[2]Department of Econometrics, School of Business and Economics, Vrije Universiteit Amsterdam, De Boelelaan 1105, 1081 HV Amsterdam, The Netherlands.
[3]Center for Research in Econometric Analysis of Time Series (CREATES), Aarhus University, Fuglesangs Allé, 4 8210 Aarhus V, Denmark

**Correspondence:** Mikkel Bennedsen (mbennedsen@econ.au.dk)

**1 Introduction**

We thank the Referee for the constructive comments on the first revision of our paper. Our replies are given below. For each item, we reproduce (a slightly paraphrased version of) the comments of the Referee, followed by our response in which we indicate how and where the paper has changed. All quoted page and line numbers ("PX, LY") refer to the newly revised version

5 of the paper, with the changes tracked.

**2 Referee report**

**2.1 Comment 1**

*First, a state-space regression using the full-year time-series need to be adressed using a Cook's distance to see how far the slope of the fit will be affected by the data sampling.*

10 ### 2.2 Response to Comment 1

We have carried out the analysis as suggested by the Referee. The analysis for detecting potential influential data points has been conducted by (1) calculating Cook's distance on the basis of sequentially leaving one observation out from the sample, and (2) estimating the slope parameter $\beta$ on the same basis, by sequentially excluding the $t$-th observation from the full sample, for $t = 1959, 1960, \ldots, 2016$. We may conclude from this two-fold analysis that none of the observations are especially influential:

15 indeed, while Cook's distance for the observation at $t = 1974$ is relatively much larger when compared to all $t \neq 1974$, it is still small in absolute terms. Similarly, the estimates of $\beta$ obtained from leaving out the $t$'th observation are very stable, also for $t = 1974$, and are in all instances well inside the $95\%$ confidence bands of the full sample estimate.

The details of these investigations are included at the end of this Response, in Figures 1–4. For completeness, we also have included this graphical output in Sect. 2.1 of the Supplementary Material file. The conclusion from these analyses (as indicated above) are mentioned in the Discussion section of the main paper (P14, L26).

**2.2.1 Changes made to the paper in response**

We have included the two analyses as indicated by (1) and (2) above: Cook's distance measures and slope parameter $\beta$ estimates in Sect. 2.1 of the Supplementary Material file. We discuss the results of these analyses (along the lines as given above) in the Discussion section of the main paper (P14, L26).

**2.3 Comment 2**

*Then, futher discussion are needed to compare trends in air-borne fraction of CO2 and in carbon sink to know/best estimates of year-to-year variability (and infered/assumed decadal variability). I mean: while your model detect a significant changes in sink rate, can we detect it in the real world given the amplitude of the year-to-decadal variability ? (please refer to the published scientific literature where possible)*

*In a situation where it doesn't, futher comparison with Gloor et al findings seems necessary.*

**2.4 Response to Comment 2**

We have extended our discussions of the estimated trends in the airborne fraction of CO2 and in the carbon sink (P14 and P15, cf. our comments below).

Our point of departure is precisely the fact that it is rather challenging to determine whether a trend is present due to the relatively large year-to-year variability.

We ask the questions whether a trend is present, whether it is deterministic (linear) or stochastic (random walk) and what the degree of measurement noise is. Our model thus resolves the high degree of year-to-year variability and allows to distinguish non-stationary processes (trend-stationary or difference-stationary) from stationary processes.

We have adopted the local linear trend model, since this basic time series model can encompass and resolve these different processes. As an additional benefit, even when the year-to-year variability in the airborne fraction and in the carbon sink originate from different processes, we have a common modeling structure.

Because year-to-year variability is high and a stochastic trend with its own source of randomness over and above the stationary measurement noise may cloud the picture further, it is desirable to use as much data as possible to detect a latent trend. It is another advantage of our model, in equation (15), that it allows to accommodate multiple data sources for the same object of interest, here the sink rate that is alternatively measured by (1) data generated from ocean and land sink models, and (2) from emission and atmospheric concentration data. Our reported estimates of the slope parameter $\beta$ are thus directly supported from two different ways to compute the sink rate.

In the case of the carbon sink rate and a single common trend for the two alternative measurements, equation (15) reduces very closely to a standard linear trend regression estimate, since the maximum likelihood estimate of the variance in the stochastic trend component is almost zero. In this case, the slope parameter $\beta$ is directly estimated from the year-to-year variation in the two different time series. Hence this result is close to the one from a standard trend regression, albeit with support from two different time series.

We believe that the reported results in our paper support the point, put forth in Gloor et al., that the behavior of the airborne fraction and the sink rate are NOT in a one-to-one correspondence. (Specifically, the point made in Gloor et al. is that findings of an increasing airborne does not necessarily imply that the sink rate is declining.) We now emphasize this more in the Discussion section (P14, L11-13 and P14, L33 – P15, L3). Note, however, that our findings also support those made by Le Quere et al., namely that the sink rate does, in fact, appear to be decreasing.

**2.4.1 Changes made to the paper in response**

The Discussion section (P14, L11-13 and P14, L33 – P15, L3) has been updated as noted above.

**2.5 Comment 3**

*Finally, regarding the model forecast: how does your model compare to a simple persistence fit (based on a first-order regression processes)?*

*I would like to stress that further discussion could be usefull in regards of the model forecast: Does your model give an idea of a possible emergence of the climate change signal from the year-to-year variability ? This would help to liase this work with the published literature on climate-carbon cycle feedbacks.*

**2.6 Response to Comment 3**

In the forecasting exercise, we now include the forecasts from a first-order autoregressive process (AR1).

The model results of the paper imply that the sink rate is linearly decreasing in time; the forecasting exercise illustrates how the sink rate will develop in the future under the assumption of the model. For instance, if the model is correct, in approximately 15 years, the sink rate will have declined below 2% (P10, L16). This has important implications for our understanding of the carbon-cycle: it indicates that the ability of the sinks to soak up $CO_2$ is decreasing, in relative terms. However, our main point is not that the sink rate will necessarily be declining with the same linear rate in the future. Our main point is that the sink rate *has been decreasing over the time period considered in the paper*. The purpose of the forecasting exercise conducted in the paper is mainly for diagnostic purposes: Does the forecasts from the model look reasonable? How can we expect the future sink rate to develop, if the model continues to be a good representation of the underlying physical processes? As such, the forecasts should not be seen as a bona fide prediction, but only as an indication of what we can expect of the sink rate, as long as it is approximately linear in concentrations (an assumption which seems to hold in the time period considered in the paper, but which might be violated in the future, cf. Appendix A in the main paper).

**2.6.1 Changes made to the paper in response**

Forecasts from an AR1 model is included in Fig. 3 (P12); some comments on this are included on P11 (L4-8).

A discussion of the forecasts and their limitations has been included in the main paper (P11, L9-15).

**3 Additional changes**

5     1. The description of the terms from Global Carbon Budget has been revised (P3).

**References**

**Figure 1.** Cook's distance of each observation from the state-space model of the airborne fraction.

[Figure]

**Figure 2.** Estimates of slope parameter $\beta$ from the state-space model of the airborne fraction after leaving out one observation. Dashed lines are 95% confidence bands for the full sample estimate.

[Figure]

**Figure 3.** Cook's distance of each observation from the state-space model of the sink rate.

[Figure]

**Figure 4.** Estimates of slope parameter $\beta$ from the state-space model of the sink rate after leaving out one observation. Dashed lines are 95% confidence bands for the full sample estimate.

[Figure]

[revised manuscript text omitted]
_2$" (Bennedsen et al., 2018): We expand on the analyses of the main paper and supply additional results.

*Copyright statement.* TEXT

**1 Introduction**

5   Section 2 provides some additional information of the univariate analyses of the main paper. In particular, Section 2.1 contains an analysis of potential statistically influential observations in the data set. Section 3 analyses data that have been averaged, so as to minimise possible influences from transitory events such as ENSO and volcanic eruptions.

**2 Additional results from univariate analyses**

In the main paper, we analyse the univariate data series in Tables 1 and 3 (airborne fraction and sink rate, respectively).
10   However, to save space, we did not provide any graphical information on the estimated trends. These are given here, in Figure 1 (airborne fraction) and Figure 2 (sink rate). The figures can be compared to the ones from the multivariate analyses, see Figures 1 and 2 in the main paper. The main take-away from these figures is that in the univariate analyses, the underlying trends are found to be time-varying. This is contrast to the multivariate analyses, where the trends are better described by a trend that does not vary in time.

15   Figure 3 plots the trend estimates of the individual sink rates, i.e., of of the ocean sink rate ($k_O$) and land sink rate ($k_L$), cf. Equation (5) in the main paper.

**Figure 1.** Univariate estimation of time-varying trend for AF.

[Figure]

[Figure]

**Figure 2.** Univariate estimation of time-varying trend for sink rate, $k_S$.

[Figure]

[Figure]

**Figure 3.** Univariate estimation of time-varying trend for land sink.

**2.1 Influential data point analysis**

To investigate the robustness of the statistical findings of the main paper, we here examine whether there are observations that are particularly influential on the analyses. We consider two related approaches. First, we calculate Cook's distance (Cook, 1977, 1979; Atkinson et al., 1997) for each $t = 1959, 1960, \ldots, 2016$, which is a measure of how influential a particular data point is. Second, we estimate the slope parameter $\beta$ after deleting the $t$'th observation from the sample (treating it as a "missing value" in the state-space system) for $t = 1959, 1960, \ldots, 2016$. That is, for each year in the sample, we obtain an estimate $\beta_{\setminus t}$ for each $t = 1959, 1960, \ldots, 2016$, by treating the $t$'th observation as missing and then estimating the slope parameter using the maximum likelihood approach considered in the main paper.

   Figures 4 and 5 present these analyses for the airborne fraction, while Figures 6 and 7 present the analogous results for the sink rate. From these figures, it is clear that one observation, namely the one at $t = 1974$, stands out: it has by far the largest Cook's distance for both the airborne fraction and the sink rate data (Figures 4 and 6, respectively). It also results in the most extreme slope parameter estimates (Figures 5 and 7, respectively). However, even this observation at $t = 1974$ seems to have only a minor influence on the analysis: indeed, the estimate of $\beta_{\setminus 1974}$ is well within the confidence bands (dashed lines) of the full sample estimate for both the airborne fraction data (Figure 5) and for the sink rate data (Figure 7). Similarly, while the Cook's distance for this data point is high relative to the other points, it is far below conventional rule-of-thumb thresholds such as $0.50$ or $1.00$, which are often used in applied analyses.

**Figure 4.** Cook's distance of each observation from the state-space model of the airborne fraction.

[Figure]

**Figure 5.** Estimates of slope parameter $\beta$ from the state-space model of the airborne fraction after leaving out one observation. Dashed lines are 95% confidence bands for the full sample estimate.

[Figure]

**Figure 6.** Cook's distance of each observation from the state-space model of the sink rate.

[Figure]

**Figure 7.** Estimates of slope parameter $\beta$ from the state-space model of the sink rate after leaving out one observation. Dashed lines are 95% confidence bands for the full sample estimate.

[Figure]

**3 Analysis with 5-year averages**

To remove possible influences of transitory events (e.g., ENSO and/or volcanic eruptions) on our analyses, we here re-do the analysis from the main paper, now with data averaged over 5 year periods.[1] We also averaged the data over 2, 3, and 4 years and found similar results to what we report below. For brevity, the results from using 2-4 year averages are not presented. They are available upon request.

We consider two ways of averaging the data: (i) Running averages, which results in overlapping "windows" of data, and (ii) non-overlapping averages. The former is considered in Section 3.1 and the latter in Section 3.2.

**3.1 Running averages**

Recall that the data of the original paper runs from $t = 1959$ to $t = 2016$, resulting in 58 observations. We now study the data after they have been averaged over a 5-year period using a running window. That is, if the original data are $x_t$, we consider the averaged data

$$\tilde{x}_t^{(m)} = \frac{1}{m} \sum_{i=1}^{m} x_{t-i+1}, \quad t = 1958 + m, 1959 + m, \ldots, 2016,$$

where $m \geq 1$ is the number of observations used in constructing the averages. For instance, if $m = 5$, we get the 5-year averages

$$\tilde{x}_t^{(5)} = \frac{1}{5} \sum_{i=1}^{5} x_{t-i+1}, \quad t = 1963, 1963, \ldots, 2016,$$

resulting in $58 - 4 = 54$ observations.

As remarked in the Discussion section of the main paper, such averaging can, unfortunately, make the error structure of the data quite complicated. We illustrate how this can happen with a toy example: Suppose that the original data are "signal plus noise":

$$x_t = x_t^* + \xi_t,$$

where $x_t^*$ is the true (unobserved) value for the underlying data and $\xi_t$ is an iid measurement error term. Let $m \geq 2$ and consider the averaged data

$$\tilde{x}_t^{(m)} = \frac{1}{m} \sum_{i=1}^{m} x_{t-i+1} = \tilde{x}_t^* + \tilde{\xi}_t,$$

where

$$\tilde{x}_t^* = \frac{1}{m} \sum_{i=1}^{m} x_{t-i+1}^*, \qquad \tilde{\xi}_t = \frac{1}{m} \sum_{i=1}^{m} \xi_{t-i+1},$$

are the averaged signal and an error term, respectively. It is clear that the error term $\tilde{\xi}_t$ is now serially correlated, which can invalidate the analysis if the researcher does not take it into account.
* * *
[1] We thank an anonymous referee for suggesting this.

With the above caveat in place, we analyse the 5-year averaged airborne fraction (AF) and sink rate (SR) data using a trend model specification, as explained in Section 3 of the main paper.[2] The results are shown in Tables 1 and 2 for the airborne fraction and Tables 3 and 4 for the sink rate. From the tables, we see that the diagnostics are quite bad, indicating that the proposed model is not able to fit the data well. In particular, we find evidence of positive serial correlation ($DW < 2$) in the
5   prediction errors, as we would expect from the discussion of the error structure above.

    We conclude that if one wants to analyse the running averages data, another approach than the one considered here is necessary. In particular, it is important to choose an approach that can take into account the error structure induced by the averaging. However, constructing such an alternative approach is outside the scope of this note. Instead, we propose to average the data using non-overlapping windows, thereby hopefully alleviating the serial correlation in the errors. The following section
10   presents the results from this approach.

**Table 1.** Univariate analysis of the airborne fraction

| | Parameter estimates | | | | | Diagnostics | | |
|---|---|---|---|---|---|---|---|---|
| | $\widehat{\sigma}_\epsilon$ | $\widehat{\sigma}_\eta$ | $\widehat{\beta}$ | s.e.$(\widehat{\beta})$ | $t$-stat$(\widehat{\beta})$ | $N$ | $R_d^2$ | $DW$ |
| $AF_t^{(1)}$ | 0 | 0.0447 | 0.00223 | 0.00584 | 0.38270 | 0.3387 | -0.0616 | 1.6657 |
| $AF_t^{(2)}$ | 0 | 0.0444 | 0.00055 | 0.00610 | 0.08987 | 1.0360 | -0.0711 | 1.8195 |

Data: 5-year running average AF. See Table 1 in the main paper for the analogous analysis on the original data.

**Table 2.** Multivariate analysis of the airborne fraction

| | Parameter estimates | | | | | Correlation matrix ($\epsilon$) | | Diagnostics | | |
|---|---|---|---|---|---|---|---|---|---|---|
| Panel A: Two individual trends as in Eq. (12) of the main paper. | | | | | | | | | | |
| | $\widehat{\sigma}_\epsilon$ | $\widehat{\sigma}_\eta$ | $\widehat{\beta}$ | s.e.$(\widehat{\beta})$ | $t$-stat$(\widehat{\beta})$ | $AF^{(1)}$ | $AF^{(2)}$ | $N$ | $R_d^2$ | $DW$ |
| $AF^{(1)}$ | 0 | 0.0425 | 0.00223 | 0.00584 | 0.38270 | 1.0000 | 0.9999 | 0.3387 | -0.0616 | 1.6657 |
| $AF^{(2)}$ | 0 | 0.0444 | 0.00055 | 0.00610 | 0.08987 | 0.9999 | 1.0000 | 1.0360 | -0.0711 | 1.8195 |
| Panel B: One common trend as in Eq. (13) of the main paper. | | | | | | | | | | |
| | $\widehat{\sigma}_\epsilon$ | $\widehat{\sigma}_\eta$ | $\widehat{\beta}$ | s.e.$(\widehat{\beta})$ | $t$-stat$(\widehat{\beta})$ | $AF^{(1)}$ | $AF^{(2)}$ | $N$ | $R_d^2$ | $DW$ |
| $AF^{(1)}$ | 0.0359 | 0.0393 | 0.00149 | 0.00540 | 0.27679 | 1.0000 | -1.0000 | 3.6409 | -0.6278 | 1.1458 |
| $AF^{(2)}$ | 0.0459 | – | – | – | – | -1.0000 | 1.0000 | 0.8089 | -0.9646 | 0.9314 |

Data: 5-year running average AF. See Table 2 in the main paper for the analogous analysis on the original data.
* * *
[2]We perform the averaging directly on the global carbon budget data, i.e., on $E_t^{ANT}, G_t, S_t^O, S_t^L$, and $C_t$, and then construct the AF and SR data as explained in Section 2 of the main paper. We also experimented with first constructing the AF and SR data and then averaging these. The results of the two approaches were very similar so we only present the results of the former.

**Table 3.** Univariate analysis of the sink rate

| | Parameter estimates | | | | | Diagnostics | | |
|---|---|---|---|---|---|---|---|---|
| | $\widehat{\sigma}_\epsilon$ | $\widehat{\sigma}_\eta$ | $\widehat{\beta}$ | s.e.($\widehat{\beta}$) | $t$-stat($\widehat{\beta}$) | $N$ | $R_d^2$ | $DW$ |
| $k_S^{(1)}$ | 0 | 0.0022 | -0.00008 | 0.00030 | -0.27290 | 4.3025 | -0.0710 | 1.7625 |
| $k_S^{(2)}$ | 0 | 0.0020 | -0.00017 | 0.00027 | -0.63234 | 0.0726 | -0.0637 | 1.7127 |

Data: 5-year running average SR. See Table 3 in the main paper for the analogous analysis on the original data.

**Table 4.** Multivariate analysis of the sink rate

| | Parameter estimates | | | | | Correlation matrix ($\epsilon$) | | Diagnostics | | |
|---|---|---|---|---|---|---|---|---|---|---|
| Panel A: Two individual trends as in Eq. (14) of the main paper. | | | | | | | | | | |
| | $\widehat{\sigma}_\epsilon$ | $\widehat{\sigma}_\eta$ | $\widehat{\beta}$ | s.e.($\widehat{\beta}$) | $t$-stat($\widehat{\beta}$) | $AF^{(1)}$ | $AF^{(2)}$ | $N$ | $R_d^2$ | $DW$ |
| $k_S^{(1)}$ | 0 | 0.0022 | -0.00008 | 0.00030 | -0.27290 | 1.0000 | 1.0000 | 4.3025 | -0.0710 | 1.7625 |
| $k_S^{(2)}$ | 0 | 0.0020 | -0.00017 | 0.00027 | -0.63235 | 1.0000 | 1.0000 | 0.0726 | -0.0637 | 1.7127 |
| Panel B: One common trend as in Eq. (15) of the main paper. | | | | | | | | | | |
| | $\widehat{\sigma}_\epsilon$ | $\widehat{\sigma}_\eta$ | $\widehat{\beta}$ | s.e.($\widehat{\beta}$) | $t$-stat($\widehat{\beta}$) | $AF^{(1)}$ | $AF^{(2)}$ | $N$ | $R_d^2$ | $DW$ |
| $k_S^{(1)}$ | 0.0024 | 0.0019 | -0.00014 | 0.00026 | -0.53638 | 1.0000 | -1.0000 | 0.6056 | -0.9981 | 0.8756 |
| $k_S^{(2)}$ | 0.0015 | – | – | – | – | -1.0000 | 1.0000 | 5.6559 | -0.6044 | 1.1971 |

Data: 5-year running average SR. See Table 4 in the main paper for the analogous analysis on the original data.

**3.2 Non-overlapping averages**

We here consider averaged data, but where the averaging is done using non-overlapping windows. That is, if $x_t$ is again the original data series, we define

$$\tilde{x}_t^{(m)} = \frac{1}{m} \sum_{i=1}^{m} x_{t-i+1}, \quad t = 1958 + m, 1958 + 2m, \dots, 2016,$$

i.e., we divide the data into bins ("windows") of size $m$ and then take the average in each bin.[3]

The advantage of this approach to averaging the data is, as discussed above, that we expect the errors to be more nicely behaved. In particular, we do not expect them to be as serially correlated in this case. The downside of the approach is of course that we will have fewer observations available for the subsequent statistical analysis. For instance, in the case of 5-year averages that we consider here, each averaged data series consists of only 12 observations.

The results are shown in Tables 5 and 6 for the airborne fraction and Tables 7 and 8 for the sink rate. From the tables, we see, as expected, that the diagnostics are better than what was seen in the case of overlapping averaging windows. Interestingly, the
* * *
[3]Because the original data set consists of 58 observations, it might not be possible in practice to have exactly $m$ observations in each bin; in this case the first bin will have less observations allocated to it. For instance, in the case of 5-year averaging, the first bin will consist of 3 observations (representing the years 1959, 1960, and 1961) and the other bins will have 5 observations allocated to them (i.e., the second bin consists of the years 1962-1966, the third bin 1967-1971, etc.).

two main conclusions from the main paper hold also in the case of the averaged data: (i) We find no statistical evidence of an increasing airborne fraction (Tables 5 and 6) but we do find statistical evidence of a decreasing sink rate (Panel B of Table 6). (ii) The latter conclusion is only reached when *all* the data are combined into one model with a common trend, compare Panel A and Panel B of Table 6.

5    Figures 8 and 9 show the trend estimates of the airborne fraction and sink rate data, respectively, in the case of averaged data using non-overlapping windows.

**Table 5.** Univariate analysis of the airborne fraction

| | Parameter estimates | | | | | Diagnostics | | |
|---|---|---|---|---|---|---|---|---|
| | $\widehat{\sigma}_\epsilon$ | $\widehat{\sigma}_\eta$ | $\widehat{\beta}$ | s.e.$(\widehat{\beta})$ | $t$-stat$(\widehat{\beta})$ | $N$ | $R^2_d$ | $DW$ |
| $AF_t^{(1)}$ | 0.0574 | 0.0067 | 0.00517 | 0.00523 | 0.98975 | 0.3594 | 0.3465 | 2.0985 |
| $AF_t^{(2)}$ | 0.0481 | 0.0450 | 0.00083 | 0.01449 | 0.05721 | 0.1227 | 0.1018 | 2.0714 |

Data: 5-year non-overlapping average AF. See Table 1 in the main paper for the analogous analysis on the original data.

**Table 6.** Multivariate analysis of the airborne fraction

| | Parameter estimates | | | | | Correlation matrix ($\epsilon$) | | Diagnostics | | |
|---|---|---|---|---|---|---|---|---|---|---|
| Panel A: Two individual trends as in Eq. (12) of the main paper. | | | | | | | | | | |
| | $\widehat{\sigma}_\epsilon$ | $\widehat{\sigma}_\eta$ | $\widehat{\beta}$ | s.e.$(\widehat{\beta})$ | $t$-stat$(\widehat{\beta})$ | $AF^{(1)}$ | $AF^{(2)}$ | $N$ | $R^2_d$ | $DW$ |
| $AF^{(1)}$ | 0.0473 | 0.0335 | 0.00538 | 0.01087 | 0.49443 | 1.0000 | 0.9456 | 0.7162 | 0.4872 | 2.5460 |
| $AF^{(2)}$ | 0.0394 | 0.0541 | 0.00122 | 0.01666 | 0.07344 | 0.9456 | 1.0000 | 0.3027 | 0.0821 | 2.3698 |
| Panel B: One common trend as in Eq. (13) of the main paper. | | | | | | | | | | |
| | $\widehat{\sigma}_\epsilon$ | $\widehat{\sigma}_\eta$ | $\widehat{\beta}$ | s.e.$(\widehat{\beta})$ | $t$-stat$(\widehat{\beta})$ | $AF^{(1)}$ | $AF^{(2)}$ | $N$ | $R^2_d$ | $DW$ |
| $AF^{(1)}$ | 0.0535 | 0 | 0.00361 | 0.00364 | 0.99164 | 1.0000 | 0.1975 | 0.4802 | 0.3698 | 2.2944 |
| $AF^{(2)}$ | 0.0596 | – | – | – | – | 0.1975 | 1.0000 | 0.7454 | 0.5965 | 2.6334 |

Data: 5-year non-overlapping average AF. See Table 2 in the main paper for the analogous analysis on the original data.

**Table 7.** Univariate analysis of the sink rate

| | Parameter estimates | | | | | Diagnostics | | |
|---|---|---|---|---|---|---|---|---|
| | $\widehat{\sigma}_\epsilon$ | $\widehat{\sigma}_\eta$ | $\widehat{\beta}$ | s.e.$(\widehat{\beta})$ | $t$-stat$(\widehat{\beta})$ | $N$ | $R^2_d$ | $DW$ |
| $k_S^{(1)}$ | 0.0023 | 0.0029 | -0.00037 | 0.00091 | -0.40483 | 0.1374 | -0.0147 | 2.0028 |
| $k_S^{(2)}$ | 0.0026 | 0.0011 | -0.00073 | 0.00041 | -1.78102 | 0.0770 | 0.2431 | 2.0450 |

Data: 5-year non-overlapping average SR. See Table 3 in the main paper for the analogous analysis on the original data.

**Table 8.** Multivariate analysis of the sink rate

| | Parameter estimates | | | | | Correlation matrix ($\epsilon$) | | Diagnostics | | |
|---|---|---|---|---|---|---|---|---|---|---|
| | $\widehat{\sigma}_\epsilon$ | $\widehat{\sigma}_\eta$ | $\widehat{\beta}$ | s.e.($\widehat{\beta}$) | $t$-stat($\widehat{\beta}$) | $AF^{(1)}$ | $AF^{(2)}$ | $N$ | $R_d^2$ | $DW$ |
| Panel A: Two individual trends as in Eq. (14) of the main paper. | | | | | | | | | | |
| $k_S^{(1)}$ | 0.0023 | 0.0029 | -0.00052 | 0.00088 | -0.58373 | 1.0000 | 1.0000 | 0.7418 | -0.0204 | 2.0524 |
| $k_S^{(2)}$ | 0.0025 | 0.0015 | -0.00078 | 0.00049 | -1.57493 | 1.0000 | 1.0000 | 0.0552 | 0.4228 | 2.2251 |
| Panel B: One common trend as in Eq. (15) of the main paper. | | | | | | | | | | |
| | $\widehat{\sigma}_\epsilon$ | $\widehat{\sigma}_\eta$ | $\widehat{\beta}$ | s.e.($\widehat{\beta}$) | $t$-stat($\widehat{\beta}$) | $AF^{(1)}$ | $AF^{(2)}$ | $N$ | $R_d^2$ | $DW$ |
| $k_S^{(1)}$ | 0.0032 | 0 | -0.00068 | 0.00020 | -3.41506 | 1.0000 | 0.3340 | 0.1292 | 0.6888 | 2.9392 |
| $k_S^{(2)}$ | 0.0027 | – | – | – | – | 0.3340 | 1.0000 | 1.0816 | 0.1869 | 1.8314 |

Data: 5-year non-overlapping average SR. See Table 4 in the main paper for the analogous analysis on the original data.

**Figure 8.** Multivariate estimation of time-varying trend for AF. Data: 5-year averages (no overlap)

**Figure 9.** Multivariate estimation of time-varying trend for sink rate, $k_S$. Data: 5-year averages (no overlap)

[Figure]

**Supplementary Material: Trend analysis of the airborne fraction and sink rate of anthropogenically released $CO_2$**

Mikkel Bennedsen[1,3], Eric Hillebrand[1,3], and Siem Jan Koopman[2,3]

[1]Department of Economics and Business Economics, Aarhus University, Fuglesangs Allé, 4 8210 Aarhus V, Denmark
[2]Department of Econometrics, School of Business and Economics, Vrije Universiteit Amsterdam, De Boelelaan 1105, 1081 HV Amsterdam, The Netherlands.
[3]Center for Research in Econometric Analysis of Time Series (CREATES), Aarhus University, Fuglesangs Allé, 4 8210 Aarhus V, Denmark

**Correspondence:** Mikkel Bennedsen (mbennedsen@econ.au.dk)

**Abstract.** This document accompanies the paper "Trend analysis of the airborne fraction and sink rate of anthropogenically released $CO_2$" (Bennedsen et al., 2018): We expand on the analyses of the main paper and supply additional results.

*Copyright statement.* TEXT

**1 Introduction**

5 Section 2 provides some additional information of the univariate analyses of the main paper.  In particular, Section 2.1 contains an analysis of potential statistically influential observations in the data set. Section 3 analyses data that have been averaged, so as to minimise possible influences from transitory events such as ENSO and volcanic eruptions.

**2 Additional results from univariate analyses**

In the main paper, we analyse the univariate data series in Tables 1 and 3 (airborne fraction and sink rate, respectively).
10 However, to save space, we did not provide any graphical information on the estimated trends. These are given here, in Figure 1 (airborne fraction) and Figure 2 (sink rate). The figures can be compared to the ones from the multivariate analyses, see Figures 1 and 2 in the main paper. The main take-away from these figures is that in the univariate analyses, the underlying trends are found to be time-varying. This is contrast to the multivariate analyses, where the trends are better described by a trend that does not vary in time.

15 Figure 3 plots the trend estimates of the individual sink rates, i.e., of of the ocean sink rate ($k_O$) and land sink rate ($k_L$), cf. Equation (5) in the main paper.

**2.1 Influential data point analysis**

**Figure 1.** Univariate estimation of time-varying trend for AF.

[Figure]

[Figure]

**Figure 2.** Univariate estimation of time-varying trend for sink rate, $k_S$.

[Figure]

[Figure]

**Figure 3.** Univariate estimation of time-varying trend for land sink.

[Figure]

To investigate the robustness of the statistical findings of the main paper, we here examine whether there are observations that are particularly influential on the analyses. We consider two related approaches. First, we calculate Cook's distance (Cook, 1977, 1979; Atkinson et al., 1997) for each $t = 1959, 1960, \ldots, 2016$, which is a measure of how influential a particular data point is. Second, we estimate the slope parameter $\beta$ after deleting the $t$'th observation from the sample (treating it as
5    a "missing value" in the state-space system) for $t = 1959, 1960, \ldots, 2016$. That is, for each year in the sample, we obtain an estimate $\beta_{\backslash t}$ for each $t = 1959, 1960, \ldots, 2016$, by treating the $t$'th observation as missing and then estimating the slope parameter using the maximum likelihood approach considered in the main paper.

     Figures 4 and 5 present these analyses for the airborne fraction, while Figures 6 and 7 present the analogous results for the sink rate. From these figures, it is clear that one observation, namely the one at $t = 1974$, stands out: it has by far the largest
10    Cook's distance for both the airborne fraction and the sink rate data (Figures 4 and 6, respectively). It also results in the most extreme slope parameter estimates (Figures 5 and 7, respectively). However, even this observation at $t = 1974$ seems to have only a minor influence on the analysis: indeed, the estimate of $\beta_{\backslash 1974}$ is well within the confidence bands (dashed lines) of the full sample estimate for both the airborne fraction data (Figure 5) and for the sink rate data (Figure 7). Similarly, while the Cook's distance for this data point is high relative to the other points, it is far below conventional rule-of-thumb thresholds
15    such as 0.50 or 1.00, which are often used in applied analyses.

[Figure]

**Figure 4.** Cook's distance of each observation from the state-space model of the airborne fraction.

[Figure]

**Figure 5.** Estimates of slope parameter $\beta$ from the state-space model of the airborne fraction after leaving out one observation. Dashed lines are 95% confidence bands for the full sample estimate.

**Figure 6.** Cook's distance of each observation from the state-space model of the sink rate.

[Figure]

**Figure 7.** Estimates of slope parameter $\beta$ from the state-space model of the sink rate after leaving out one observation. Dashed lines are 95% confidence bands for the full sample estimate.

[Figure]

**3 Analysis with 5-year averages**

To remove possible influences of transitory events (e.g., ENSO and/or volcanic eruptions) on our analyses, we here re-do the analysis from the main paper, now with data averaged over 5 year periods.[1] We also averaged the data over 2, 3, and 4 years and found similar results to what we report below. For brevity, the results from using 2-4 year averages are not presented. They are available upon request.

We consider two ways of averaging the data: (i) Running averages, which results in overlapping "windows" of data, and (ii) non-overlapping averages. The former is considered in Section 3.1 and the latter in Section 3.2.

**3.1 Running averages**

Recall that the data of the original paper runs from $t = 1959$ to $t = 2016$, resulting in 58 observations. We now study the data after they have been averaged over a 5-year period using a running window. That is, if the original data are $x_t$, we consider the averaged data

$$\tilde{x}_t^{(m)} = \frac{1}{m} \sum_{i=1}^{m} x_{t-i+1}, \quad t = 1958 + m, 1959 + m, \ldots, 2016,$$

where $m \geq 1$ is the number of observations used in constructing the averages. For instance, if $m = 5$, we get the 5-year averages

$$\tilde{x}_t^{(5)} = \frac{1}{5} \sum_{i=1}^{5} x_{t-i+1}, \quad t = 1963, 1963, \ldots, 2016,$$

resulting in $58 - 4 = 54$ observations.

As remarked in the Discussion section of the main paper, such averaging can, unfortunately, make the error structure of the data quite complicated. We illustrate how this can happen with a toy example: Suppose that the original data are "signal plus noise":

$$x_t = x_t^* + \xi_t,$$

where $x_t^*$ is the true (unobserved) value for the underlying data and $\xi_t$ is an iid measurement error term. Let $m \geq 2$ and consider the averaged data

$$\tilde{x}_t^{(m)} = \frac{1}{m} \sum_{i=1}^{m} x_{t-i+1} = \tilde{x}_t^* + \tilde{\xi}_t,$$

where

$$\tilde{x}_t^* = \frac{1}{m} \sum_{i=1}^{m} x_{t-i+1}^*, \qquad \tilde{\xi}_t = \frac{1}{m} \sum_{i=1}^{m} \xi_{t-i+1},$$

are the averaged signal and an error term, respectively. It is clear that the error term $\tilde{\xi}_t$ is now serially correlated, which can invalidate the analysis if the researcher does not take it into account.
* * *
[1] We thank an anonymous referee for suggesting this.

With the above caveat in place, we analyse the 5-year averaged airborne fraction (AF) and sink rate (SR) data using a trend model specification, as explained in Section 3 of the main paper.[2] The results are shown in Tables 1 and 2 for the airborne fraction and Tables 3 and 4 for the sink rate. From the tables, we see that the diagnostics are quite bad, indicating that the proposed model is not able to fit the data well. In particular, we find evidence of positive serial correlation ($DW < 2$) in the prediction errors, as we would expect from the discussion of the error structure above.

We conclude that if one wants to analyse the running averages data, another approach than the one considered here is necessary. In particular, it is important to choose an approach that can take into account the error structure induced by the averaging. However, constructing such an alternative approach is outside the scope of this note. Instead, we propose to average the data using non-overlapping windows, thereby hopefully alleviating the serial correlation in the errors. The following section presents the results from this approach.

**Table 1.** Univariate analysis of the airborne fraction

| | Parameter estimates | | | | | Diagnostics | | |
|---|---|---|---|---|---|---|---|---|
| | $\widehat{\sigma}_\epsilon$ | $\widehat{\sigma}_\eta$ | $\widehat{\beta}$ | s.e.$(\widehat{\beta})$ | $t$-stat$(\widehat{\beta})$ | $N$ | $R_d^2$ | $DW$ |
| $AF_t^{(1)}$ | 0 | 0.0447 | 0.00223 | 0.00584 | 0.38270 | 0.3387 | -0.0616 | 1.6657 |
| $AF_t^{(2)}$ | 0 | 0.0444 | 0.00055 | 0.00610 | 0.08987 | 1.0360 | -0.0711 | 1.8195 |

Data: 5-year running average AF. See Table 1 in the main paper for the analogous analysis on the original data.

**Table 2.** Multivariate analysis of the airborne fraction

| | Parameter estimates | | | | | Correlation matrix ($\epsilon$) | | Diagnostics | | |
|---|---|---|---|---|---|---|---|---|---|---|
| Panel A: Two individual trends as in Eq. (12) of the main paper. | | | | | | | | | | |
| | $\widehat{\sigma}_\epsilon$ | $\widehat{\sigma}_\eta$ | $\widehat{\beta}$ | s.e.$(\widehat{\beta})$ | $t$-stat$(\widehat{\beta})$ | $AF^{(1)}$ | $AF^{(2)}$ | $N$ | $R_d^2$ | $DW$ |
| $AF^{(1)}$ | 0 | 0.0425 | 0.00223 | 0.00584 | 0.38270 | 1.0000 | 0.9999 | 0.3387 | -0.0616 | 1.6657 |
| $AF^{(2)}$ | 0 | 0.0444 | 0.00055 | 0.00610 | 0.08987 | 0.9999 | 1.0000 | 1.0360 | -0.0711 | 1.8195 |
| Panel B: One common trend as in Eq. (13) of the main paper. | | | | | | | | | | |
| | $\widehat{\sigma}_\epsilon$ | $\widehat{\sigma}_\eta$ | $\widehat{\beta}$ | s.e.$(\widehat{\beta})$ | $t$-stat$(\widehat{\beta})$ | $AF^{(1)}$ | $AF^{(2)}$ | $N$ | $R_d^2$ | $DW$ |
| $AF^{(1)}$ | 0.0359 | 0.0393 | 0.00149 | 0.00540 | 0.27679 | 1.0000 | -1.0000 | 3.6409 | -0.6278 | 1.1458 |
| $AF^{(2)}$ | 0.0459 | – | – | – | – | -1.0000 | 1.0000 | 0.8089 | -0.9646 | 0.9314 |

Data: 5-year running average AF. See Table 2 in the main paper for the analogous analysis on the original data.
* * *
[2]We perform the averaging directly on the global carbon budget data, i.e., on $E_t^{ANT}, G_t, S_t^O, S_t^L$, and $C_t$, and then construct the AF and SR data as explained in Section 2 of the main paper. We also experimented with first constructing the AF and SR data and then averaging these. The results of the two approaches were very similar so we only present the results of the former.

**Table 3.** Univariate analysis of the sink rate

| | Parameter estimates | | | | | Diagnostics | | |
|---|---|---|---|---|---|---|---|---|
| | $\widehat{\sigma}_\epsilon$ | $\widehat{\sigma}_\eta$ | $\widehat{\beta}$ | s.e.$(\widehat{\beta})$ | $t$-stat$(\widehat{\beta})$ | $N$ | $R_d^2$ | $DW$ |
| $k_S^{(1)}$ | 0 | 0.0022 | -0.00008 | 0.00030 | -0.27290 | 4.3025 | -0.0710 | 1.7625 |
| $k_S^{(2)}$ | 0 | 0.0020 | -0.00017 | 0.00027 | -0.63234 | 0.0726 | -0.0637 | 1.7127 |

Data: 5-year running average SR. See Table 3 in the main paper for the analogous analysis on the original data.

**Table 4.** Multivariate analysis of the sink rate

| | Parameter estimates | | | | | Correlation matrix ($\epsilon$) | | Diagnostics | | |
|---|---|---|---|---|---|---|---|---|---|---|
| Panel A: Two individual trends as in Eq. (14) of the main paper. | | | | | | | | | | |
| | $\widehat{\sigma}_\epsilon$ | $\widehat{\sigma}_\eta$ | $\widehat{\beta}$ | s.e.$(\widehat{\beta})$ | $t$-stat$(\widehat{\beta})$ | $AF^{(1)}$ | $AF^{(2)}$ | $N$ | $R_d^2$ | $DW$ |
| $k_S^{(1)}$ | 0 | 0.0022 | -0.00008 | 0.00030 | -0.27290 | 1.0000 | 1.0000 | 4.3025 | -0.0710 | 1.7625 |
| $k_S^{(2)}$ | 0 | 0.0020 | -0.00017 | 0.00027 | -0.63235 | 1.0000 | 1.0000 | 0.0726 | -0.0637 | 1.7127 |
| Panel B: One common trend as in Eq. (15) of the main paper. | | | | | | | | | | |
| | $\widehat{\sigma}_\epsilon$ | $\widehat{\sigma}_\eta$ | $\widehat{\beta}$ | s.e.$(\widehat{\beta})$ | $t$-stat$(\widehat{\beta})$ | $AF^{(1)}$ | $AF^{(2)}$ | $N$ | $R_d^2$ | $DW$ |
| $k_S^{(1)}$ | 0.0024 | 0.0019 | -0.00014 | 0.00026 | -0.53638 | 1.0000 | -1.0000 | 0.6056 | -0.9981 | 0.8756 |
| $k_S^{(2)}$ | 0.0015 | – | – | – | – | -1.0000 | 1.0000 | 5.6559 | -0.6044 | 1.1971 |

Data: 5-year running average SR. See Table 4 in the main paper for the analogous analysis on the original data.

**3.2 Non-overlapping averages**

We here consider averaged data, but where the averaging is done using non-overlapping windows. That is, if $x_t$ is again the original data series, we define

$$\tilde{x}_t^{(m)} = \frac{1}{m} \sum_{i=1}^{m} x_{t-i+1}, \quad t = 1958 + m, 1958 + 2m, \ldots, 2016,$$

i.e., we divide the data into bins ("windows") of size $m$ and then take the average in each bin.[3]

The advantage of this approach to averaging the data is, as discussed above, that we expect the errors to be more nicely behaved. In particular, we do not expect them to be as serially correlated in this case. The downside of the approach is of course that we will have fewer observations available for the subsequent statistical analysis. For instance, in the case of 5-year averages that we consider here, each averaged data series consists of only 12 observations.

The results are shown in Tables 5 and 6 for the airborne fraction and Tables 7 and 8 for the sink rate. From the tables, we see, as expected, that the diagnostics are better than what was seen in the case of overlapping averaging windows. Interestingly, the
* * *
[3]Because the original data set consists of 58 observations, it might not be possible in practice to have exactly $m$ observations in each bin; in this case the first bin will have less observations allocated to it. For instance, in the case of 5-year averaging, the first bin will consist of 3 observations (representing the years 1959, 1960, and 1961) and the other bins will have 5 observations allocated to them (i.e., the second bin consists of the years 1962-1966, the third bin 1967-1971, etc.).

two main conclusions from the main paper hold also in the case of the averaged data: (i) We find no statistical evidence of an increasing airborne fraction (Tables 5 and 6) but we do find statistical evidence of a decreasing sink rate (Panel B of Table 6). (ii) The latter conclusion is only reached when *all* the data are combined into one model with a common trend, compare Panel A and Panel B of Table 6.

5    Figures 8 and 9 show the trend estimates of the airborne fraction and sink rate data, respectively, in the case of averaged data using non-overlapping windows.

**Table 5.** Univariate analysis of the airborne fraction

| | Parameter estimates | | | | | Diagnostics | | |
|---|---|---|---|---|---|---|---|---|
| | $\widehat{\sigma}_\epsilon$ | $\widehat{\sigma}_\eta$ | $\widehat{\beta}$ | s.e.$(\widehat{\beta})$ | $t$-stat$(\widehat{\beta})$ | $N$ | $R_d^2$ | $DW$ |
| $AF_t^{(1)}$ | 0.0574 | 0.0067 | 0.00517 | 0.00523 | 0.98975 | 0.3594 | 0.3465 | 2.0985 |
| $AF_t^{(2)}$ | 0.0481 | 0.0450 | 0.00083 | 0.01449 | 0.05721 | 0.1227 | 0.1018 | 2.0714 |

Data: 5-year non-overlapping average AF. See Table 1 in the main paper for the analogous analysis on the original data.

**Table 6.** Multivariate analysis of the airborne fraction

| | Parameter estimates | | | | | Correlation matrix ($\epsilon$) | | Diagnostics | | |
|---|---|---|---|---|---|---|---|---|---|---|
| Panel A: Two individual trends as in Eq. (12) of the main paper. | | | | | | | | | | |
| | $\widehat{\sigma}_\epsilon$ | $\widehat{\sigma}_\eta$ | $\widehat{\beta}$ | s.e.$(\widehat{\beta})$ | $t$-stat$(\widehat{\beta})$ | $AF^{(1)}$ | $AF^{(2)}$ | $N$ | $R_d^2$ | $DW$ |
| $AF^{(1)}$ | 0.0473 | 0.0335 | 0.00538 | 0.01087 | 0.49443 | 1.0000 | 0.9456 | 0.7162 | 0.4872 | 2.5460 |
| $AF^{(2)}$ | 0.0394 | 0.0541 | 0.00122 | 0.01666 | 0.07344 | 0.9456 | 1.0000 | 0.3027 | 0.0821 | 2.3698 |
| Panel B: One common trend as in Eq. (13) of the main paper. | | | | | | | | | | |
| | $\widehat{\sigma}_\epsilon$ | $\widehat{\sigma}_\eta$ | $\widehat{\beta}$ | s.e.$(\widehat{\beta})$ | $t$-stat$(\widehat{\beta})$ | $AF^{(1)}$ | $AF^{(2)}$ | $N$ | $R_d^2$ | $DW$ |
| $AF^{(1)}$ | 0.0535 | 0 | 0.00361 | 0.00364 | 0.99164 | 1.0000 | 0.1975 | 0.4802 | 0.3698 | 2.2944 |
| $AF^{(2)}$ | 0.0596 | – | – | – | – | 0.1975 | 1.0000 | 0.7454 | 0.5965 | 2.6334 |

Data: 5-year non-overlapping average AF. See Table 2 in the main paper for the analogous analysis on the original data.

**Table 7.** Univariate analysis of the sink rate

| | Parameter estimates | | | | | Diagnostics | | |
|---|---|---|---|---|---|---|---|---|
| | $\widehat{\sigma}_\epsilon$ | $\widehat{\sigma}_\eta$ | $\widehat{\beta}$ | s.e.$(\widehat{\beta})$ | $t$-stat$(\widehat{\beta})$ | $N$ | $R_d^2$ | $DW$ |
| $k_S^{(1)}$ | 0.0023 | 0.0029 | -0.00037 | 0.00091 | -0.40483 | 0.1374 | -0.0147 | 2.0028 |
| $k_S^{(2)}$ | 0.0026 | 0.0011 | -0.00073 | 0.00041 | -1.78102 | 0.0770 | 0.2431 | 2.0450 |

Data: 5-year non-overlapping average SR. See Table 3 in the main paper for the analogous analysis on the original data.

**Table 8.** Multivariate analysis of the sink rate

| | Parameter estimates | | | | | Correlation matrix ($\epsilon$) | | Diagnostics | | |
|---|---|---|---|---|---|---|---|---|---|---|
| | $\widehat{\sigma}_\epsilon$ | $\widehat{\sigma}_\eta$ | $\widehat{\beta}$ | s.e.($\widehat{\beta}$) | $t$-stat($\widehat{\beta}$) | $AF^{(1)}$ | $AF^{(2)}$ | $N$ | $R_d^2$ | $DW$ |
| Panel A: Two individual trends as in Eq. (14) of the main paper. | | | | | | | | | | |
| $k_S^{(1)}$ | 0.0023 | 0.0029 | -0.00052 | 0.00088 | -0.58373 | 1.0000 | 1.0000 | 0.7418 | -0.0204 | 2.0524 |
| $k_S^{(2)}$ | 0.0025 | 0.0015 | -0.00078 | 0.00049 | -1.57493 | 1.0000 | 1.0000 | 0.0552 | 0.4228 | 2.2251 |
| Panel B: One common trend as in Eq. (15) of the main paper. | | | | | | | | | | |
| $k_S^{(1)}$ | 0.0032 | 0 | -0.00068 | 0.00020 | -3.41506 | 1.0000 | 0.3340 | 0.1292 | 0.6888 | 2.9392 |
| $k_S^{(2)}$ | 0.0027 | – | – | – | – | 0.3340 | 1.0000 | 1.0816 | 0.1869 | 1.8314 |

Data: 5-year non-overlapping average SR. See Table 4 in the main paper for the analogous analysis on the original data.

**Figure 8.** Multivariate estimation of time-varying trend for AF. Data: 5-year averages (no overlap)

**Figure 9.** Multivariate estimation of time-varying trend for sink rate, $k_S$. Data: 5-year averages (no overlap)

[Figure]

**Correspondence:** Mikkel Bennedsen (mbennedsen@econ.au.dk)

**1 Introduction**

We thank the Referee for the constructive comments on the first revision of our paper. Our replies are given below. For each item, we reproduce (a slightly paraphrased version of) the comments of the Referee, followed by our response in which we indicate how and where the paper has changed. All quoted page and line numbers ("PX, LY") refer to the newly revised version of the paper, with the changes tracked.

**2 Referee report**

**2.1 Comment 1**

*First, a state-space regression using the full-year time-series need to be adressed using a Cook's distance to see how far the slope of the fit will be affected by the data sampling.*

**2.2 Response to Comment 1**

We have carried out the analysis as suggested by the Referee. The analysis for detecting potential influential data points has been conducted by (1) calculating Cook's distance on the basis of sequentially leaving one observation out from the sample, and (2) estimating the slope parameter $\beta$ on the same basis, by sequentially excluding the $t$-th observation from the full sample, for $t = 1959, 1960, \ldots, 2016$. We may conclude from this two-fold analysis that none of the observations are especially influential: indeed, while Cook's distance for the observation at $t = 1974$ is relatively much larger when compared to all $t \neq 1974$, it is still small in absolute terms. Similarly, the estimates of $\beta$ obtained from leaving out the $t$'th observation are very stable, also for $t = 1974$, and are in all instances well inside the $95\%$ confidence bands of the full sample estimate.

The details of these investigations are included at the end of this Response, in Figures 1–4. For completeness, we also have included this graphical output in Sect. 2.1 of the Supplementary Material file. The conclusion from these analyses (as indicated above) are mentioned in the Discussion section of the main paper (P14, L26).

**2.2.1 Changes made to the paper in response**

5   We have included the two analyses as indicated by (1) and (2) above: Cook's distance measures and slope parameter $\beta$ estimates in Sect. 2.1 of the Supplementary Material file. We discuss the results of these analyses (along the lines as given above) in the Discussion section of the main paper (P14, L26).

**2.3 Comment 2**

*Then, futher discussion are needed to compare trends in air-borne fraction of CO2 and in carbon sink to know/best estimates*
10   *of year-to-year variability (and infered/assumed decadal variability). I mean: while your model detect a significant changes in sink rate, can we detect it in the real world given the amplitude of the year-to-decadal variability ? (please refer to the published scientific literature where possible)*

  *In a situation where it doesn't, futher comparison with Gloor et al findings seems necessary.*

**2.4 Response to Comment 2**

15   We have extended our discussions of the estimated trends in the airborne fraction of CO2 and in the carbon sink (P14 and P15, cf. our comments below).

  Our point of departure is precisely the fact that it is rather challenging to determine whether a trend is present due to the relatively large year-to-year variability.

  We ask the questions whether a trend is present, whether it is deterministic (linear) or stochastic (random walk) and what the
20   degree of measurement noise is. Our model thus resolves the high degree of year-to-year variability and allows to distinguish non-stationary processes (trend-stationary or difference-stationary) from stationary processes.

  We have adopted the local linear trend model, since this basic time series model can encompass and resolve these different processes. As an additional benefit, even when the year-to-year variability in the airborne fraction and in the carbon sink originate from different processes, we have a common modeling structure.

25   Because year-to-year variability is high and a stochastic trend with its own source of randomness over and above the stationary measurement noise may cloud the picture further, it is desirable to use as much data as possible to detect a latent trend. It is another advantage of our model, in equation (15), that it allows to accommodate multiple data sources for the same object of interest, here the sink rate that is alternatively measured by (1) data generated from ocean and land sink models, and (2) from emission and atmospheric concentration data. Our reported estimates of the slope parameter $\beta$ are thus directly supported from
30   two different ways to compute the sink rate.

In the case of the carbon sink rate and a single common trend for the two alternative measurements, equation (15) reduces very closely to a standard linear trend regression estimate, since the maximum likelihood estimate of the variance in the stochastic trend component is almost zero. In this case, the slope parameter $\beta$ is directly estimated from the year-to-year variation in the two different time series. Hence this result is close to the one from a standard trend regression, albeit with support from two different time series.

We believe that the reported results in our paper support the point, put forth in Gloor et al., that the behavior of the airborne fraction and the sink rate are NOT in a one-to-one correspondence. (Specifically, the point made in Gloor et al. is that findings of an increasing airborne does not necessarily imply that the sink rate is declining.) We now emphasize this more in the Discussion section (P14, L11-13 and P14, L33 – P15, L3). Note, however, that our findings also support those made by Le Quere et al., namely that the sink rate does, in fact, appear to be decreasing.

**2.4.1 Changes made to the paper in response**

The Discussion section (P14, L11-13 and P14, L33 – P15, L3) has been updated as noted above.

**2.5 Comment 3**

*Finally, regarding the model forecast: how does your model compare to a simple persistence fit (based on a first-order regression processes)?*

*I would like to stress that further discussion could be usefull in regards of the model forecast: Does your model give an idea of a possible emergence of the climate change signal from the year-to-year variability ? This would help to liase this work with the published literature on climate-carbon cycle feedbacks.*

**2.6 Response to Comment 3**

In the forecasting exercise, we now include the forecasts from a first-order autoregressive process (AR1).

The model results of the paper imply that the sink rate is linearly decreasing in time; the forecasting exercise illustrates how the sink rate will develop in the future under the assumption of the model. For instance, if the model is correct, in approximately 15 years, the sink rate will have declined below 2% (P10, L16). This has important implications for our understanding of the carbon-cycle: it indicates that the ability of the sinks to soak up $CO_2$ is decreasing, in relative terms. However, our main point is not that the sink rate will necessarily be declining with the same linear rate in the future. Our main point is that the sink rate *has been decreasing over the time period considered in the paper*. The purpose of the forecasting exercise conducted in the paper is mainly for diagnostic purposes: Does the forecasts from the model look reasonable? How can we expect the future sink rate to develop, if the model continues to be a good representation of the underlying physical processes? As such, the forecasts should not be seen as a bona fide prediction, but only as an indication of what we can expect of the sink rate, as long as it is approximately linear in concentrations (an assumption which seems to hold in the time period considered in the paper, but which might be violated in the future, cf. Appendix A in the main paper).

**2.6.1 Changes made to the paper in response**

Forecasts from an AR1 model is included in Fig. 3 (P12); some comments on this are included on P11 (L4-8).

A discussion of the forecasts and their limitations has been included in the main paper (P11, L9-15).

**3 Additional changes**

5    1. The description of the terms from Global Carbon Budget has been revised (P3).

**References**

**Figure 1.** Cook's distance of each observation from the state-space model of the airborne fraction.

[Figure]

**Figure 2.** Estimates of slope parameter $\beta$ from the state-space model of the airborne fraction after leaving out one observation. Dashed lines are 95% confidence bands for the full sample estimate.

[Figure]

**Figure 3.** Cook's distance of each observation from the state-space model of the sink rate.

[Figure]

**Figure 4.** Estimates of slope parameter $\beta$ from the state-space model of the sink rate after leaving out one observation. Dashed lines are 95% confidence bands for the full sample estimate.

[Figure]